# Model Guidance via Robust Feature Attribution

**Mihnea Ghitu**  *mihnea.ghitu20@imperial.ac.uk*
*Imperial College London*

**Vihari Piratla**  *viharipiratla@gmail.com*
*University of Cambridge*

**Matthew Wicker**  *m.wicker@imperial.ac.uk*
*Imperial College London*

**Reviewed on OpenReview:** *https://openreview.net/forum?id=AVAHxDSqUu*

## Abstract

Controlling the patterns a model learns is essential to preventing reliance on irrelevant or misleading features. Such reliance on irrelevant features, often called shortcut features, has been observed across domains, including medical imaging and natural language processing, where it may lead to real-world harms. A common mitigation strategy leverages annotations (provided by humans or machines) indicating which features are relevant or irrelevant. These annotations are compared to model explanations, typically in the form of feature salience, and used to guide the loss function during training. Unfortunately, recent works have demonstrated that feature salience methods are unreliable and therefore offer a poor signal to optimize. In this work, we propose a simplified objective that simultaneously optimizes for explanation robustness and mitigation of shortcut learning. Unlike prior objectives with similar aims, we demonstrate theoretically why our approach ought to be more effective. Across a comprehensive series of experiments, we show that our approach consistently reduces test-time misclassifications by 20% compared to state-of-the-art methods. We also extend prior experimental settings to include natural language processing tasks. Additionally, we conduct novel ablations that yield practical insights, including the relative importance of annotation quality over quantity. Code for our method and experiments is available at: https://github.com/Mihneaghitu/ModelGuidanceViaRobustFeatureAttribution.

## 1 Introduction

Machine learning (ML) has seen tremendous progress in the last decade, culminating in mainstream adoption in a variety of domains. Deep neural networks (DNN) sit at the core of this progress, due to their remarkable performance and ease of deployment (Goodfellow et al., 2016). With applications in wide-ranging domains (Bommasani et al., 2021), the promise of impact comes with the potential for substantial harm when deploying models that may be dependent on irrelevant and misleading feature patterns present in their particular training dataset (Ross et al., 2017).

A model misgeneralizing due to its reliance on incidental correlations is often referred to as *shortcut learning* (Ross et al., 2017; Geirhos et al., 2020; Heo et al., 2023). Leveraging such shortcuts during training significantly hinders the reliable deployment of models in safety-critical domains. For instance, models were documented to exploit dataset-specific incidental correlations such as hospital tags in chest x-rays when diagnosing pneumonia (Zech et al., 2018; DeGrave et al., 2021) or bandage in skin lesions when diagnosising skin cancer (Rieger et al., 2020). In natural language processing, models often rely incorrectly on pronouns and proper nouns when making decisions (McCoy et al., 2019), leading to bias in downstream tasks such as hiring and loan approvals (Rudinger et al., 2018). Left unchecked, learning and relying upon such shortcuts may lead to unintended harms at deployment time. Thus, understanding and mitigating shortcut learning has become

an important area of research that has accumulated rich benchmarks and methodologies (Ross et al., 2017; Geirhos et al., 2020; Singla et al., 2022).

The best-performing approaches to mitigating shortcut learning involves the use of per-input feature-level annotations that indicate if a feature in a given input is a *core* feature or a *non-core* feature that may represent a shortcut (e.g., the background of an image) (Heo et al., 2023). Throughout the paper we will refer to non-core features simply as *masked* features. Methods that make use of such annotations are commonly termed Machine Learning from Explanations (MLX) methods. A variety of MLX approaches have been proposed (Lee et al., 2022; Heo et al., 2023), and a common theme across state-of-the-art methods is the use of model explanations to regularize dependence on masked features, pioneered in *right for the right reasons* (Ross et al., 2017). The model explanations used in MLX methods come in the form of feature attribution explanations which assign a score to each feature indicating how important the score is to the model prediction. By ensuring that masked features have low importance (achieved via regularizing the loss function) MLX methods mitigate shortcut learning (Jia et al., 2018; Kavumba et al., 2021; Heo et al., 2023; Ross et al., 2017; Jia et al., 2018; Kavumba et al., 2021; Heo et al., 2023).

However, these methods assume that feature attribution explanations provide a faithful signal of model behavior, an assumption increasingly called into question. Recent work shows that the most popular form of feature importance, gradient-based attributions, can be unstable and easily manipulated, even without changing model predictions (Dombrowski et al., 2019; Wicker et al.). This fragility stems from the non-linear nature of deep networks, which allows imperceptible input changes to dramatically alter gradient-based explanations—much like adversarial attacks on outputs (Goodfellow et al., 2014). As a result, using only the gradient at a single input point in conjunction with annotation masks may not reliably suppress shortcut learning because out-of-distribution data might render the explanations unusable. Moreover, prior MLX works largely overlook how annotation quality and availability affect performance and inference-time generalizability, leaving important practical considerations unexplored.

In this work, we address these shortcomings by first proposing that mitigating reliance on shortcut features would be substantially more effective if one additionally optimizes for the reliability of the feature saliency explanations. We propose a robust variant of right for right reasons (RRR) that accounts for the noise in feature importance scores, which we dub *robustly right for the right reasons* ($R^4$). $R^4$ regularizes the feature importance in a high-dimensional ball around the training points rather than regularizing the feature importance at the training point alone. As a result, the objective required an inner optimization over a high-dimensional manifold, which is intractable. Thus, we instantiate three variants of $R^4$ each implementing a different approximate solution to the optimization problem: *Rand*-$R^4$ takes the maximum over perturbations sampled from the domain of the optimization, *Adv*-$R^4$ uses first-order optimization to search the domain of the optimization, and *Cert*-$R^4$ uses convex-relaxations of the problem to compute an upper-bound the solution to the optimization problem. We compare each of these methods with a series of state-of-the-art MLX approaches with various datasets. We employ two synthetic datasets: the well-studied DecoyMNIST dataset (Ross et al., 2017) as well as a novel shortcut learning benchmark derived from a medical image diagnosis problem (Yang et al., 2023). Through further evaluation on three real-world benchmarks, we demonstrate consistent gains of $R^4$ over the state-of-the-art by a substantial margin (a raw increase of 6% accuracy on average). In summary, this paper makes the following contributions:

- We establish a novel, adversarial approach to machine learning from explanations that we name *robustly right for the right reasons* or $R^4$.

- We propose and implement three (approximate) solutions to the intractable inner optimization of $R^4$: a statistical approach, a first-order optimization approach, and a convex-relaxation based approach and demonstrate their scaling to models as large as ResNet-18 and BERT.

- Through a series of experiments, we demonstrate that $R^4$ outperforms prior state of the art MLX approaches across all datasets and can effectively mitigate shortcut learning with annotations on just 20% of the examples in the training set.

## 2  Related Works

**Shortcut Learning.** The phenomena of models learning non-generalizing dataset artifacts is referred to as *shortcut learning* and is well-studied (Geirhos et al., 2020; Shah et al., 2020). Mitigating shortcut learning has been approached from multiple fronts, the three major themes are: (a) annotating training examples with group identity (Sagawa et al.; Ye et al., 2024) to delineate examples with positive and negative incidental correlation, (b) diversifying the training set bias-free or bias nullifying examples (Lee et al., 2022), learning from explicit annotation of relevant and irrelevant parts of input called *machine learning from explanations* (MLX) (Ross et al., 2017; Rieger et al., 2020; Heo et al., 2023). MLX is the focus of our work, which enables specifying the irrelevant features explicitly (Sagawa et al.; Ye et al., 2024; Ross et al., 2017). Additionally, a line of works augments the provided human explanations with natural language (Selvaraju et al., 2019) or through iterative interaction (Schramowski et al., 2020; Linardatos et al., 2020).

**Machine Learning from Explanations.** MLX approaches operate through an augmented loss objective that regularizes the model's dependence on irrelevant features. Traditionally, an input feature attribution method is used to compute the importance of various input features. Previous proposals differed greatly in their choice of the feature attribution method: Ross et al. (2017) employed gradient-based feature explanation, Rieger et al. (2020) employed contextual decomposition-based feature explanation (Singh et al., 2018), Schramowski et al. (2020) employed LIME (Ribeiro et al., 2016), Shao et al. (2021) employed influence functions (Koh & Liang, 2017). Heo et al. (2023) championed directly minimizing the function's sensitivity to irrelevant feature perturbations. In Zhang et al. (2024) the authors propose to learn feature importance for CNNs and mitigate shortcut reliance via regularization of intermediate activations. As is evident from the focus of many previous work, the core difficulty lies in approximating the importance of input features.

**Fragility of Explanations.** Feature attribution explanations are among the most popular explanation methods and include shapely values (Lundberg & Lee, 2017), LIME (Ribeiro et al., 2016), smooth- and integrated-gradients (Smilkov et al., 2017; Sundararajan et al., 2017), grad-CAM (Selvaraju et al., 2020), among others Leofante & Wicker (2025). While each of these methods aim at capturing the per-feature importance of a model decision with respect to a given input, they have recently been found to give opposite feature importance scores for indistinguishable inputs (Dombrowski et al., 2019). Manipulating explanations by slightly perturbing input values is similar to crafting adversarial examples (Goodfellow et al., 2014), thus adversarial training techniques have been adopted and employed to enforce the robustness of explanations (Dombrowski et al., 2022; Wicker et al.). However, current MLX approaches do not enforce robustness of explanations and thus use an unreliable optimization signal. Moreover, adopting prior works such as Wicker et al. cannot be done directly due to the complications of the MLX setting, that is, the existence of masked features.

**Robustness and MLX.** The work most related to ours is Heo et al. (2023), which championed for directly imposing robustness to $\epsilon$-ball perturbations of irrelevant features per example. The methods presented in our work instead explore robustness of explanations, the signal that is optimized, as opposed to robustness of predictions. Despite their resemblance, $R^4$ is more elegant with only one term added to the objective while also being empirically superior. We further provide theoretical insights (§ 4) and empirical validation (§ 5) on $R^4$'s merit over Heo et al. (2023).

## 3  Preliminaries

We denote a machine learning model as a parametric function $f$ with parameters $\boldsymbol{\theta} \in \mathbb{R}^m$, which maps from features $\boldsymbol{x} \in \mathbb{R}^n$ to labels $\boldsymbol{y} \in \mathcal{Y}$. We consider supervised learning in the classification setting with a labeled dataset $\mathcal{D} = \{(\boldsymbol{x}^{(i)}, \boldsymbol{y}^{(i)})\}_{i=1}^N$. In the MLX setting, in addition to the feature-label pairs, we have access to human or machine-provided explanations in the form of masks $\boldsymbol{m}^{(i)} \in \mathbb{R}^n$, which highlight the important regions or components of the input features relevant for the prediction. These masks provide a form of weak supervision, guiding the model's attention during training. Thus, in MLX, the dataset is extended to include these annotations and is represented as $\mathcal{D}_{\mathrm{MLX}} = \{(\boldsymbol{x}^{(i)}, \boldsymbol{y}^{(i)}, \boldsymbol{m}^{(i)})\}_{i=1}^K$, where each $\boldsymbol{m}^{(i)}$ is the explanation associated with the $i$-th data point. In particular, the mask $\boldsymbol{m}^{(i)}$ is a vector with entries in the interval $[0, 1]$, where 1 represents fully irrelevant features and 0 represents fully relevant features.

**Right for the right reasons.** The seminal approach to MLX, right for the right reasons (RRR or $R^3$) proposes to modify the standard loss by including a regularizing term to suppress gradient values for irrelevant features as shown below.

$$\mathcal{L}_{\mathrm{RRR}}(\boldsymbol{\theta}) = \underbrace{\ell(f^{\boldsymbol{\theta}}(\boldsymbol{x}^{(i)}), \boldsymbol{y}^{(i)})}_{\text{right answer}} + \lambda \underbrace{\|\boldsymbol{m}^{(i)} \odot \nabla_{\boldsymbol{x}} f^{\boldsymbol{\theta}}(\boldsymbol{x}^{(i)})\|_2^2}_{\text{right reason}} + \beta \underbrace{\|\boldsymbol{\theta}\|_2^2}_{\text{regularize}} \tag{1}$$

In the above expression, $\odot$ denotes the Hadamard product. The loss function $\mathcal{L}_{\mathrm{RRR}}$ involves three components. The first, labeled "right answer," is the standard loss used to train the model. The second, labeled "right reason", penalizes the magnitude of the gradient along the feature dimensions determined to be irrelevant, hence the element-wise multiplication with the mask $\boldsymbol{m}^{(i)}$. The final term, labeled "regularize," is the standard weight decay term that has a smoothing effect on the final model. $\lambda$ and $\beta$ are hyperparameters that control the relative contributions of the gradient penalty and the weight decay term, respectively.

**Input-gradient robustness.** The critical term in $R^3$ is the "right reason" component: $\|\boldsymbol{m}^{(i)} \odot \nabla_{\boldsymbol{x}} f^{\boldsymbol{\theta}}(\boldsymbol{x}^{(i)})\|_2^2$, which depends on the input gradient $\nabla_{\boldsymbol{x}} f^{\boldsymbol{\theta}}(\boldsymbol{x}^{(i)})$ to ensure that the model does not leverage information from irrelevant features to make predictions. However, the ineffectiveness of gradient explanations for highly non-linear models is well-documented (Heo et al., 2023; Wicker et al.), owing to their poor robustness to perturbations of spurious features. (Wicker et al.) employed a variant of Lipschitz smoothness to quantify the robustness of an explanation method through input gradient *fragility*. The fragility ($\delta$) of a function with parameters $\boldsymbol{\theta}$ in an $\epsilon$-ball around $x$ is defined as:

$$\forall \boldsymbol{x}' \in \mathcal{B}_\epsilon(\boldsymbol{x}), \|\nabla_{\boldsymbol{x}} f^{\boldsymbol{\theta}}(\boldsymbol{x}) - \nabla_{\boldsymbol{x}'} f^{\boldsymbol{\theta}}(\boldsymbol{x}')\| \leq \delta \tag{2}$$

We can understand this intuitively by observing that as $\delta \to 0$ we require that the gradients become identical for all inputs in the ball and therefore the model is linear inside of $B_\epsilon(\boldsymbol{x})$. On the other hand, when $\delta \to \infty$ we can interpret this as the model becoming more and more non-linear. Unfortunately, even for small convolutional neural networks trained without regularization, typical values of $\delta$ will have extreme magnitudes (Table 2), denoting that even an imperceptible perturbation ($\leq \epsilon$) in the irrelevant features can drastically alter the feature importance, thus making it an unreliable feature to optimize.

# 4  $R^4$ : Robustly Right for the Right Reasons

The primary motivation for our methodology is the observation that when the input gradient is non-robust the $R^3$ regularizer is optimizing a poor signal (Dombrowski et al., 2019; 2022) and can be substantially improved if one simultaneously minimizes the $R^3$ loss and $\delta$. To do so, we propose an adversarial learning objective that we call *robustly right for the right reasons* or $R^4$.

$$\mathcal{L}_{R^4}(\boldsymbol{\theta}) = \underbrace{\ell(f^{\boldsymbol{\theta}}(\boldsymbol{x}^{(i)}), \boldsymbol{y}^{(i)})}_{\text{right answer}} + \lambda \underbrace{\max_{\boldsymbol{\xi}:\|\boldsymbol{\xi}\|<\epsilon} \|\boldsymbol{m}^{(i)} \odot \nabla_{\boldsymbol{x}} f^{\boldsymbol{\theta}}(\boldsymbol{x}^{(i)} + \boldsymbol{m}^{(i)} \odot \boldsymbol{\xi})\|_2}_{\text{robustly right reason}} \tag{3}$$

The inner optimization (the maximum) is the adversarial term that simultaneously enforces that the gradient magnitude of irrelevant features is small and remains small for any small change to the irrelevant features. Unfortunately, computing the solution to this maximization problem is intractable for complex models and even for fully-connected networks is NP-Complete (Katz et al., 2017), thus approximate solutions must be proposed in order to use this objective in practice. In the following sections, we present the theoretical motivation for $R^4$ in §4.1 then provide a series of approximations for intractable inner optimization §4.2.1 - §4.2.3. The extension of the proposed approximations to language modeling tasks can be found in Appendix C.

## 4.1  $R^4$ Theoretical Intuition

We begin by describing the theoretical intuition of our learning objective, showing its benefits relative to prior works. We denote the set of all perturbations of an input $x'$ in the features determined by the mask $m$ with magnitude at most $\epsilon$ to be $\mathcal{B}_\epsilon^m(\boldsymbol{x}')$ and highlight that an ideal model is insensitive to changes in the

input in this set for moderate to large values of $\epsilon$. To think about the behavior of the model in this set we employ a second-order Taylor expansion around $x'$ in the direction determined by $m$:

$$f_{\boldsymbol{m},2}^{\boldsymbol{\theta}}(\boldsymbol{x}) \approx f^{\boldsymbol{\theta}}(\boldsymbol{x}') + \underbrace{\nabla_{\boldsymbol{x}} f^{\boldsymbol{\theta}}(\boldsymbol{x}')^{\top}(\boldsymbol{m} \odot (\boldsymbol{x} - \boldsymbol{x}'))}_{\text{function change}} + \underbrace{\frac{1}{2}(\boldsymbol{m} \odot (\boldsymbol{x} - \boldsymbol{x}'))^{\top} H_{\boldsymbol{x}'}(f^{\boldsymbol{\theta}})(\boldsymbol{m} \odot (\boldsymbol{x} - \boldsymbol{x}'))}_{\text{gradient change}}, \qquad (4)$$

where the subscript 2 in $f_{\boldsymbol{m},2}^{\theta}$ denotes the order of the approximation, and where $H_{x'}(f^{\theta})$ denotes the Hessian of the function induced by a model with parameters $\theta$, computed with respect to the input and evaluated at $x'$. We define the function $f_{\boldsymbol{1-m},2}^{\boldsymbol{\theta}}(\boldsymbol{x})$ correspondingly to denote the function's behavior when perturbing features not in $m$. The complete approximation of the function can then be expressed as:

$$f^{\boldsymbol{\theta}}(\boldsymbol{x}) \approx \underbrace{f_{\boldsymbol{m},2}^{\boldsymbol{\theta}}(\boldsymbol{x})}_{\text{sens. to shortcuts}} + \underbrace{f_{\boldsymbol{1-m},2}^{\boldsymbol{\theta}}(\boldsymbol{x})}_{\text{sens. to core}} - f^{\boldsymbol{\theta}}(\boldsymbol{x}'). \qquad (5)$$

The primary goal of the $\text{R}^4$ algorithm can be stated as minimizing only the contribution of $f_{\boldsymbol{m},2}^{\boldsymbol{\theta}}(\boldsymbol{x})$. To observe this, we assume a bound on the gradient magnitude for any point in $\mathcal{B}_{\epsilon}^{m}(\boldsymbol{x}')$:

$$\forall \boldsymbol{x}^{\star} \in \mathcal{B}_{\epsilon}^{m}(\boldsymbol{x}'), \|\boldsymbol{m} \odot \nabla_{\boldsymbol{x}} f^{\boldsymbol{\theta}}(\boldsymbol{x}^{\star})\| \leq \delta^{\star} \qquad (6)$$

By rearranging the Taylor expansion $f_{\boldsymbol{m},2}^{\boldsymbol{\theta}}(\boldsymbol{x})$, applying the norm on both sides and using the triangle inequality we can use Equation 6 to obtain:

$$\|f_{\boldsymbol{m},2}^{\boldsymbol{\theta}}(\boldsymbol{x}) - f_{\boldsymbol{m},2}^{\boldsymbol{\theta}}(\boldsymbol{x}')\| \lesssim \delta^{\star}\|\boldsymbol{x} - \boldsymbol{x}'\| + \frac{1}{2}\|\boldsymbol{x} - \boldsymbol{x}'\|\|H_{\boldsymbol{x}'}(f^{\boldsymbol{\theta}})\|\|\boldsymbol{x} - \boldsymbol{x}'\|$$

We have by definition that $\boldsymbol{x}' \in \mathcal{B}_{\epsilon}^{m}(\boldsymbol{x})$, which implies $\|\boldsymbol{x} - \boldsymbol{x}'\| \leq \|\boldsymbol{m} \odot \epsilon\| \leq \|\epsilon\|$, since $m \in [0,1]$. In addition, the norm of the input gradient at $x'$ is bounded by $\delta^{\star}$ in an $\epsilon$-ball then the norm of the Hessian at $x'$ is bounded by $\delta^{\star}$. Additionally, using Taylor's Theorem with Lagrange remainder, we have that the bound in Equation 6 allows us to make the above inequality strict regardless of the order of the approximation. Therefore, we have:

$$\|f_{m}^{\boldsymbol{\theta}}(\boldsymbol{x}) - f_{m}^{\boldsymbol{\theta}}(\boldsymbol{x}')\| \leq \delta^{\star}\|\epsilon\|(1 + \frac{1}{2}\|\epsilon\|) \qquad (7)$$

where the contribution from all higher-order terms is captured by $\delta^{\star}\|\epsilon\|^{2}/2$. Full proof can be found in Appendix H. The primary intuition for $\text{R}^4$ is that $\epsilon$ is a moderate to large value (as it acts only on the masked values $m$) and $\delta^{\star}$ can be extremely large for complex models §(3). Since $\text{R}^4$ directly minimizes the bound in Equation 6, we can see how we effectively minimize the sensitivity of the function to perturbations of the masked features. We highlight that though we suppress the change in $f_{m}^{\boldsymbol{\theta}}$ we leave the function $f_{1-m}^{\boldsymbol{\theta}}$ unregularized thus encouraging models to learn from core features while suppressing spurious features. This differs from the mechanisms employed in prior works (Ross et al., 2017; Heo et al., 2023), which require weight regularization to achieve SOTA results.

**Shortcomings of $\text{R}^3$ (Ross et al., 2017).** Heo et al. (2017) have shown that $\text{R}^3$ is suboptimal because it requires heavy parameter regularization (third term of Equation 1) to minimize the contribution of spurious features. Additionally, suppressing the parameter norm in $\text{R}^3$ also has the effect of minimizing $\delta$. However, since parameter smoothing is agnostic to feature saliency, regularizing the model, while minimizing the input fragility and contribution of spurious features, will also significantly hamper learning from core features.

**Shortcomings of IBP-Ex + $\text{R}^3$ (Heo et al., 2023).** The approach proposed by Heo et al. (2023), IBP-Ex+$\text{R}^3$, employs adversarial methods to minimize the change in output of the model, which serves to partially suppress use of spurious features (the $\delta^{\star}\|\epsilon\|$ in our bound). However, they rely on $\text{R}^3$ to minimize the *gradient change* contribution. As we have established this overlooks the higher-order terms that contribute the practically non-negligible term: $\delta^{\star}\|\epsilon\|^{2}/2$ and therefore leads to a sub-optimal mitigation of shortcut learning. In addition to our argument here, we report theoretical analysis in the same spirit as Heo et al. (2023) to demonstrate the advantage of $\text{R}^4$ over IBP-Ex + $\text{R}^3$ in Appendix I.

## 4.2 Inner Optimization Approximations

We propose three approximation strategies for tackling the inner maximization problem in equation 3: **Rand-$\mathbf{R}^4$**, **Adv-$\mathbf{R}^4$** and **Cert-$\mathbf{R}^4$**. The first leverages gradient sampling in the masked region under perturbations of the annotated features, offering scalability to large neural network architectures. The second employs a projected gradient descent–based search, providing a trade-off between scalability and tightness of the approximation. The third utilizes interval bound propagation (IBP) (Gowal et al., 2018) to obtain worst-case predictions under manipulations of the shortcut features, yielding the tightest bounds, but suffering from scalability limitations due to overapproximation errors, especially for large networks, as demonstrated in our experiments. For our discussion of computational runtime, we consider a test set with $N$ points, and denote the time required for a standard forward pass as $T_f$, and for a backward pass as $T_b$, values which are dependent on the type of model architecture used.

### 4.2.1 Rand-$R^4$: A statistical approach

The first approach we present to approximate the intractable inner maximization of $R^4$ is a scalable statistical approach based on sampling. The intuition for this approach is that our optimization is over all possible perturbations of the input in the masked region. Thus, a loose but efficient approximation of the maximum is to randomly sample perturbations and take the worst sample as the estimate of the maximum. Where we define $\mathcal{U}(x, m, \epsilon)$ as the uniform distribution over perturbations of $x$ in the region determined by $m$ with magnitude at most $\epsilon$, the approach that we will call Rand-$R^4$ is given by:

$$r_{(i+1)} = \max\{r_{(i)}, ||\nabla_{\boldsymbol{x}} f^{\boldsymbol{\theta}}(\boldsymbol{x}) - \nabla_{\boldsymbol{x}_{(i)}} f^{\boldsymbol{\theta}}(\boldsymbol{x}_{(i)})||\}, \ \boldsymbol{x}_{(i)} \sim \mathcal{U}(\boldsymbol{x}, \boldsymbol{m}, \epsilon),$$

where $r_0 = 0$. This approach is run for a fixed number of samples $k$ and then the value $r_k$ is taken to be the approximate solution to the $R^4$ optimization problem. Naturally, the probability of sampling the optimal solution is 0, however, as $k \to \infty$ we expect to get close to the optimal solution. In practice, this approach offers a good balance between computational complexity and result quality; its strength lies not in its tightness but in its ease of implementation, scalability, and adaptability. While the method can become time-consuming for large sample sizes, fewer samples are often sufficient for large datasets, and the approach lends itself well to parallelization. In terms of runtime, we can notice that for every data point, a perturbation sample requires a forward and backward pass in order to obtain the gradient. Therefore, the total runtime is $N \times k \times (T_f + T_b)$.

### 4.2.2 Adv-$R^4$: An adversarial attack approach

The learning objective that we propose in this work closely resembles that of adversarial training for enhancing robustness machine learning models. There are, however, two important distinctions that make traditional adversarial training approaches incompatible with our objective. Firstly, in standard adversarial training *all* features can be perturbed while in our objective only features in the masked region $\boldsymbol{m}$ can be perturbed. To accommodate this change, one can perform projected gradient descent (PGD) with the projection being onto the set $B^m_\epsilon(\boldsymbol{x})$. Additionally, and more importantly, standard adversarial attacks target changing the model's prediction/output. In contrast, our objective is to penalize the maximum change in the input gradients. This is similar to what is proposed in the robust explanations literature (Dombrowski et al., 2019); however, that line of work does not consider the constraints imposed by the MLX setting. Considering all the modifications suggested by our approach as a whole, we have that a locally optimal solution to the optimization problem in $R^4$ can be found by $k$ iterations of the following scheme, letting $\boldsymbol{x}^{\mathrm{adv}}_{(0)} = \boldsymbol{x}$:

$$\boldsymbol{x}_{(i+1)} = \boldsymbol{x}^{\mathrm{adv}}_{(i)} + \alpha \mathrm{sgn}\left(\nabla_{\boldsymbol{x}}|\nabla_{\boldsymbol{x}} f^{\boldsymbol{\theta}}(\boldsymbol{x}) - \nabla_{\boldsymbol{x}} f^{\boldsymbol{\theta}}(\boldsymbol{x}^{\mathrm{adv}}_{(i)})|\right)$$

$$\boldsymbol{x}^{\mathrm{adv}}_{(i+1)} = \mathrm{Proj}\left(\boldsymbol{x}_{(i+1)}, B^{\boldsymbol{m}}_\epsilon(\boldsymbol{x})\right),$$

where $\mathrm{sgn}(\cdot)$ is the sign function, $\mathrm{Proj}(\cdot, \cdot)$ is the projection operator and $\alpha$ is a hyperparameter representing the step size of a general PGD attack Madry et al. (2017), controlling the magnitude of each perturbation step. The result $\boldsymbol{x}^{\mathrm{adv}}_{(k)}$ is then an input point which approximately maximizes the inner optimization of

$R^4$. Unfortunately, owing to the non-convexity of the inner optimization $R^4$, this approach will always under-estimate the true solution to the optimization problem.

The benefits of this approach to the $R^4$ learning objective that adversarial attacks are well-studied, and thus there are many known heuristics for jointly improving model performance and adversarial performance. Unfortunately, it is well-known that models with large capacity tend to learn the patterns within the attacks themselves (Madry et al., 2017; Dong et al., 2022). For safety critical domains such as medical image classification, overfitting to specific attack types may not be an acceptable approximation (Tramer et al., 2020). The runtime of this method grows as the number of iterations increases, requiring two forward and backward passes for each PGD iteration. As such, the total runtime is $N \times 2 \times k \times (T_f + T_b)$.

### 4.2.3 Cert-$R^4$: Convex relaxation approach

Both Rand-$R^4$ and Adv-$R^4$ are efficient, but only provide a lower bound on the maximization problem of interest. Although a lower bound on $R^4$ may suffice to avoid shortcut learning, in safety-critical domains such as autonomous navigation and medical imaging, approximately avoiding shortcuts may not be sufficient to prevent adversarial behavior, such as relying on unknown spurious features or reacting unpredictably to imperceptible input changes. Thus, it is critical to provide practitioners in these domains with tools that offer *worst-case guarantees* against *any* type of adversarial behavior. In this section, we present Cert-$R^4$ which leverages advances in convex relaxation for neural networks to upper-bound the maximization in the $R^4$ objective, which corresponds to the bound in Equation 6 in our theoretical discussion. We emphasize that the fact that we compute an upper bound in this case enables the strict inequality in Equation equation 7. Here, we describe how one can use interval bound propagation to over-approximate the $R^4$ learning objective.

Though the $R^4$ objective can be applied to any differentiable model, we focus on neural networks. We begin by defining a neural network model $f^{\boldsymbol{\theta}} : \mathbb{R}^{n_{\text{in}}} \to \mathbb{R}^{n_{\text{out}}}$ with $K$ layers and parameters $\boldsymbol{\theta} = \left\{ (\boldsymbol{W}^{(i)}, \boldsymbol{b}^{(i)}) \right\}_{i=1}^{K}$ as:

$$\hat{\boldsymbol{z}}^{(k)} = \boldsymbol{W}^{(k)} z^{(k-1)} + \boldsymbol{b}^{(k)},$$
$$\boldsymbol{z}^{(k)} = \sigma \left( \hat{\boldsymbol{z}}^{(k)} \right)$$

where $\boldsymbol{z}^{(0)} = \boldsymbol{x}$, $f^{\boldsymbol{\theta}}(\boldsymbol{x}) = \hat{\boldsymbol{z}}^{(K)}$, and $\sigma$ is the activation function, which we assume is monotonic. We also state the backwards pass here starting with $\boldsymbol{d}^{(L)} = \nabla_{\hat{\boldsymbol{z}}^{(L)}} f^{\boldsymbol{\theta}}(\boldsymbol{x})$, we have that backwards pass is given by:

$$\boldsymbol{d}^{(k-1)} = \left( \boldsymbol{W}^{(k)} \right)^{\top} \boldsymbol{d}^{(k)} \odot \sigma' \left( \hat{\boldsymbol{z}}^{(k-1)} \right)$$

where we are interested in $\boldsymbol{d}^{(0)} = \nabla_{\boldsymbol{x}} f^{\boldsymbol{\theta}}(\boldsymbol{x})$. The important observation for the above equations (both forwards and backwards) is that they require only matrix multiplication, addition, and the application of a monotonic non-linearity. As such, we can efficiently employ interval arithmetic to compute all possible values of $\delta^{(0)}$ by first casting the domain of the optimization problem as an interval: $[\boldsymbol{x}^L, \boldsymbol{x}^U]$ where $\boldsymbol{x}^L = \boldsymbol{x} - \epsilon \boldsymbol{m}$ and $\boldsymbol{x}^U = \boldsymbol{x} + \epsilon \boldsymbol{m}$ for some positive constant $\epsilon$. In Wicker et al. a procedure using interval bounds is given that takes such an interval over inputs and computes an interval over gradients $[\boldsymbol{\eta}^L, \boldsymbol{\eta}^U]$ such that we have the following property:

$$\forall \boldsymbol{x}' \in [\boldsymbol{x} - \epsilon \boldsymbol{m}, \boldsymbol{x} + \epsilon \boldsymbol{m}], \ \nabla_{\boldsymbol{x}'} f^{\boldsymbol{\theta}}(\boldsymbol{x}') \in [\boldsymbol{\eta}^L, \boldsymbol{\eta}^U]. \tag{8}$$

We provide an account of how this propagation proceeded in Appendix J. To demonstrate the use of interval bound propagation to the $R^4$ objective, we first compute the interval over input gradients, $[\boldsymbol{\eta}^L, \boldsymbol{\eta}^U]$ and now show how we can compute, in closed form, the solution to a maximum upper-bounding the inner maximization of $R^4$:

$$\boldsymbol{\eta}_i^{\star} = \begin{cases} \boldsymbol{\eta}_i^L & \text{if } |\nabla_{\boldsymbol{x}} f^{\boldsymbol{\theta}}(\boldsymbol{x})_i - \boldsymbol{\eta}_i^L| > |\nabla_{\boldsymbol{x}} f^{\boldsymbol{\theta}}(\boldsymbol{x})_i - \boldsymbol{\eta}_i^U| \\ \boldsymbol{\eta}_i^U & \text{otherwise} \end{cases}$$

Finally, the value $|\boldsymbol{\eta}^{\star} - \nabla_{\boldsymbol{x}} f^{\boldsymbol{\theta}}(\boldsymbol{x})|$ is an upper-bound on the maximum of the $R^4$ objective. Unfortunately, due to the fact that $[\boldsymbol{\eta}^L, \boldsymbol{\eta}^U]$ computed using interval bound will always be loose, thus the term $|\boldsymbol{\eta}^{\star} - \nabla_{\boldsymbol{x}} f^{\boldsymbol{\theta}}(\boldsymbol{x})|$

Table 1: Performance Comparison of MLX Methods. The best results, along with those not statistically distinguishable from the best, are highlighted in bold. Values following the $\pm$ symbol indicate standard deviations.

| Learning Objective ↓ | Synthetic Datasets | | | | | |
|---|---|---|---|---|---|---|
| | Decoy MNIST | | Decoy DERM | | Decoy IMDB | |
| | Avg Acc | Wg Acc | Avg Acc | Wg Acc | Avg Acc | Wg Acc |
| ERM | $63.89 \pm 1.2$ | $22.39 \pm 1.7$ | $50.32 \pm 0.0$ | $48.39 \pm 0.0$ | $50.02 \pm 0.0$ | $0.1 \pm 0.0$ |
| $R^3$ | $92.22 \pm 0.7$ | $85.43 \pm 5.9$ | $68.44 \pm 1.7$ | $68.32 \pm 8.8$ | $71.43 \pm 2.27$ | $64.72 \pm 19.66$ |
| Smooth-$R^3$ | $93.31 \pm 0.19$ | $85.28 \pm 1.9$ | $70.95 \pm 2.4$ | $70.78 \pm 7.1$ | $73.96 \pm 3.76$ | $68.61 \pm 25.25$ |
| IBP-Ex | $89.69 \pm 0.5$ | $83.65 \pm 4.2$ | $68.85 \pm 1.1$ | $67.47 \pm 9.6$ | - | - |
| IBP-Ex + $R^3$ | $93.07 \pm 0.1$ | $88.05 \pm 1.9$ | $70.15 \pm 0.9$ | $70.13 \pm 5.6$ | - | - |
| Rand-$R^4$ | $93.29 \pm 0.17$ | $88.92 \pm 1.4$ | $71.13 \pm 1.2$ | $69.47 \pm 0.4$ | $78.17 \pm 3.61$ | $73.97 \pm 5.36$ |
| Adv-$R^4$ | $93.47 \pm 0.2$ | $89.51 \pm 1.2$ | $72.26 \pm 0.6$ | $70.47 \pm 1.3$ | $\mathbf{86.8 \pm 2.68}$ | $\mathbf{83.02 \pm 12.26}$ |
| Cert-$R^4$ | $\mathbf{97.02 \pm 0.09}$ | $\mathbf{94.70 \pm 0.4}$ | $\mathbf{78.24 \pm 0.6}$ | $\mathbf{78.11 \pm 4.9}$ | - | - |

| Learning Objective ↓ | Real World Datasets | | | | | |
|---|---|---|---|---|---|---|
| | ISIC | | Plant | | Salient Imagenet | |
| | Avg Acc | Wg Acc | Avg Acc | Wg Acc | Avg Acc | RCS |
| ERM | $82.02 \pm 0.05$ | $51.66 \pm 3.1$ | $70.83 \pm 1.9$ | $52.53 \pm 10.0$ | $99.10$ | $48.48$ |
| $R^3$ | $72.53 \pm 0.9$ | $63.00 \pm 4.06$ | $74.53 \pm 6.2$ | $64.80 \pm 20.6$ | $95.53$ | $51.56$ |
| Smooth-$R^3$ | $86.17 \pm 1.4$ | $64.59 \pm 10.5$ | $69.83 \pm 9.9$ | $61.86 \pm 24.53$ | $96.42$ | $53.42$ |
| IBP-Ex | $82.81 \pm 1.7$ | $69.62 \pm 11.7$ | $75.82 \pm 6.4$ | $72.44 \pm 21.2$ | - | - |
| IBP-Ex + $R^3$ | $83.33 \pm 1.1$ | $70.91 \pm 2.1$ | $76.48 \pm 1.6$ | $75.97 \pm 5.08$ | - | - |
| Rand-$R^4$ | $\mathbf{87.28 \pm 1.8}$ | $\mathbf{78.15 \pm 4.5}$ | $67.23 \pm 8.2$ | $66.45 \pm 29.19$ | $98.21$ | $62.43$ |
| Adv-$R^4$ | $85.65 \pm 1.2$ | $66.65 \pm 1.9$ | $79.58 \pm 2.4$ | $79.20 \pm 5.7$ | $\mathbf{99.10}$ | $\mathbf{68.51}$ |
| Cert-$R^4$ | $\mathbf{85.39 \pm 1.3}$ | $\mathbf{71.52 \pm 9.2}$ | $\mathbf{82.72 \pm 2.3}$ | $\mathbf{82.25 \pm 1.08}$ | - | - |

will always be *larger* than the true maximum. We note that if the upper-bound computed by Cert-$R^4$ is sufficiently small, then by the argument provided in our theoretical discussion we have that neither the models prediction nor its input gradient changes in response to changes in the masked region. The computational requirements of this method are the smallest out of all our proposed approximations, since for every point only two forward and backward passes are necessary. Therefore, the total runtime is $N \times 2 \times (T_f + T_b)$.

# 5 Experiments

We conduct experiments on six datasets, comparing each $R^4$ variant against baselines, including state-of-the-art MLX methods. Appendix A contains a detailed description of the dataset characteristics and feature annotations, while Appendix B outlines the model architectures used in our experiments. The datasets include three synthetic ones and three real-world datasets: **Decoy MNIST** (Ross et al., 2017), **Decoy DERM** (a variant of DermMNIST (Yang et al., 2023) created similarly to Decoy MNIST), **Decoy IMDB** (a text dataset (Maas et al., 2011) created by mimicking Decoy MNIST in a discrete space), **ISIC** (Codella et al., 2019), **Plant Phenotyping**, and **Salient ImageNet** (Singla et al., 2022). Additional results can be found in Appendix D, E and F.

## 5.1 Baseline Algorithms

**Empirical Risk Minimization (ERM).** We consider the standard empirical risk minimization (ERM) as our simplest benchmark. Without any regularization, ERM simply minimizes the categorical-cross entropy

loss using the Adam optimizer. This is equivalent to simply using the term represented by the "*right answer*" term in equation 3.

**Regularization-based methods.** We describe the traditional approaches to MLX as *regularization-based* and take the primary benchmark for this category to be $R^3$ as described in Ross et al. (2017) and discussed in Section 3. In order to understand if simply adopting a different, more robust, explanation method is sufficient to overcome the lack of robustness of input-gradient information we additionally employ Smooth-$R^3$, which adopts the same loss as $R^3$, but employs the smoothed gradient of Smilkov et al. (2017) as the explanation method that identifies reliance on shortcut features.

**Robustness-based methods.** To benchmark against the recently proposed robustness-based methods, we employ IBP-Ex (Heo et al., 2023) and use IBP-Ex+$R^3$ which was found to have state-of-the-art performance across all datasets (we note their paper refers to the latter as IBP-Ex+Grad-Reg).

## 5.2 Performance Metrics

Our objective is to suppress the influence of irrelevant features while preserving overall predictive performance. To evaluate this, we report two complementary metrics: worst-group accuracy (**Wg Acc**), which reflects the effectiveness of irrelevant feature suppression, and macro-averaged group accuracy (**Avg Acc**), which captures overall model performance across groups. These metrics are computed over predefined groups specific to each dataset and have also been used by the previous SOTA method of Heo et al. (2023). For Salient-ImageNet, we utilize the relative core sensitivity (RCS) (Singla et al., 2022) which measures the function's sensitivity to perturbations of the core features and compares it to the function's sensitivity to perturbations of masked features. In order to understand if our hypothesis about the relationship between robustness of input gradients and effectively regularizing shortcut features is correct, we also consider the certified fragility of input gradients in the masked regions which we denote with $\kappa$ (Wicker et al.); this is the average difference between certified upper and lower-bounds of the input gradient of all masked features.

## 5.3 Avoiding Shortcut Learning with MLX

Table 1 summarizes the performance of various state-of-the-art (SOTA) gradient regularization techniques across our six benchmark datasets. We report average accuracy (Avg Acc), worst-group accuracy (Wg Acc), as well as their respective standard deviations measured across runs. For SalientImageNet and DecoyIMDB, we omit results for Cert-$R^4$, IBP-Ex, and IBP-Ex+$R^4$, as these methods rely on IBP, which exhibits limited scalability to large-scale architectures such as ResNet-18 and BERT. The overall results indicate a clear advantage of robustness-based methods, particularly those employing interval bound propagation, in mitigating shortcut learning across diverse datasets. The Cert-$R^4$ model consistently ranks among the top-performing methods which achieve accuracy better than previous SOTA: IBP-Ex+$R^3$.

Although computationally cheap, gradient-based methods such as Smooth-$R^3$ show improvements over standard empirical risk minimization, while hybrid interval-and-gradient-based techniques such as IBP-Ex+$R^3$ achieve the best accuracy among previous existing methods. Our method not only improves upon these results, but also reduces computational overhead—thanks to the strength of our regularization objective—by enabling by enabling the use of more computationally efficient variants such as Adv-$R^4$. These findings suggest that in safety-critical and high-resolution image classification tasks our method is a powerful approach to overcome shortcut reliance and enhancing model reliability, while maintaining a low computational overhead, if desired.

We highlight that in all but one metric across Decoy-MNIST, Decoy-DERM, ISIC, and Plant the Cert-$R^4$ method performs best in terms of macro-average group accuracy and worst-group accuracy, suggesting that Cert-$R^4$ is effective in mitigating shortcut learning. We note that on Plant, the gains from Cert-R4 are influenced by IBP's overapproximation errors and the presence of large saliency regions covering a significant area of each image, which limit the relative benefit of its strong regularization. This effect is less pronounced in datasets with sparser explanations or smaller saliency regions, where Cert-R4 achieves the greatest improvement.

### 5.4 Can masked feature suppresion encourage core feature learning?

Table 2: Input Gradient Fragility Ratio (Masked/Core).

| | Dataset | | | |
|---|---|---|---|---|
| | Decoy MNIST | Decoy DERM | ISIC | Plant |
| **Learning Objective** $\downarrow$ | $\kappa_m/\kappa_{1-m}$ $(\kappa_m)(\downarrow)$ | | | |
| ERM | 1.497 (5.0e2) | 0.851 (6.7e3) | 0.933 (4.2e-1) | 0.909 (4.0e3) |
| $R^3$ | 0.565 (4.4) | 0.992 (5.8e3) | **0.835** (1.7e-3) | 0.970 (4.5e3) |
| Smooth-$R^3$ | 0.570 (4.3) | 0.995 (3.9e3) | 0.879 (1.0) | 1.023 (2.1e3) |
| IBP-Ex | 0.645 (5.6) | 0.802 (3.0e3) | 0.929 (1.8) | 0.971 (4.7e3) |
| IBP-Ex + $R^3$ | 0.536 (3.4) | 0.785 (2.9e3) | 0.941 (2.9) | 0.904 (1.1e3) |
| Rand-$R^4$ | 0.339 (2.1) | 0.997 (3.8e3) | 0.869 (1.4) | 0.984 (1.3e3) |
| Adv-$R^4$ | 0.321 (2.0) | 0.985 (3.5e3) | 0.889 (4.0e-1) | **0.733** (1.8e3) |
| Cert-$R^4$ | **0.024** (9.3e-1) | **0.003** (1.3) | 0.842 (9.0e-1) | 0.914 (8.9e1) |

In order to ensure that our proposed method fares better in suppressing the contribution of masked features, while encouraging learning from core features, we compute the ratio between the input gradient *fragility* in the masked ($\kappa_m$) and core ($\kappa_{1-m}$) regions of the input. A low ratio and small $\kappa_m$ value indicate reduced input-gradient fragility (i.e., improved robustness) and reflect a model's improved ability to prioritize core features over non-core ones during learning, reflecting model sensitivity as discussed in §4.1. We report this metric as $\kappa_{1-m}/\kappa_m$ in Table 2, and show the absolute value of $\kappa_m$ in blue to indicate the order of magnitude. Mathematically, this can be written as $\|\boldsymbol{\eta_m^L} - \boldsymbol{\eta_m^U}\| \leq \kappa_m$ and $\|\boldsymbol{\eta_{1-m}^L} - \eta_{1-m}^U\| \leq \kappa_{1-m}$, where $\boldsymbol{\eta_m^L}, \boldsymbol{\eta_m^U}, \boldsymbol{\eta_{1-m}^L}, \boldsymbol{\eta_{1-m}^U}$ are defined as in property 8, with the only difference being that the latter two are computed with respect to the core features (i.e. masks are inverted).

From Table 2, we observe that Cert-$R^4$ significantly outperforms previous MLX approaches in DecoyMNIST and DecoyDERM, being **14** and, respectively, **276** times better than the best performing previous technique. Much lower ratio scores demonstrate that, in contrast to prior methods, our approach simultaneously discourages masked feature reliance, while encouraging learning from core regions. Lastly, we see that in the case of ISIC, a noisy, partially masked dataset, our method only competes with $R^3$ in terms of this metric. However, judging from the absolute $\kappa_m$ value, we notice that $R^3$ is insensitive to both core and masked features, which means that it learns poorly from *any* region, fact evident in Table 1's **Wg Acc** as well. Thus, this observation establishes Cert-R4 as the most performant approach on ISIC in terms of relative robustness.

### 5.5 Exploring the effect of noisy and partially specified masks

In real-world scenarios, datasets are likely to contain few and potentially noisy masks, as obtaining high-quality annotations is often costly. Therefore, we aim to evaluate how effectively our method can learn shortcut feature patterns under such conditions, relative to baseline approaches. This provides a more informative benchmark of how MLX techniques perform in realistic, non-ideal settings. Accordingly, in Figure 1, we present a series of ablations examining the sample complexity of state-of-the-art MLX methods. Our findings are complemented by additional ablations of other critical hyperparameters, such as model size in Appendix E, and of the effect of mask corruptions on previous SOTA methods ($R^3$ and IBP-Ex) in Appendix F. We hope this analysis serves as a foundation for developing more sample-efficient approaches, which are particularly relevant in resource-constrained or safety-critical domains.

**Resilience to data and mask variations.** In the left column of Figure 1 we plot the results of simultaneously reducing the amount of data and masks (e.g., we keep the original proportion of masks in the dataset fixed). The general trend we observe indicates that while traditional regularization methods ($R^3$) struggle with limited mask *and* data availability, robustness-based techniques are far more effective in maintaining worst-group accuracy. In particular, we notice that our novel method's statistical and adversarial variants fare better at maintaining effective in sustaining worst-group accuracy even in a low-data regime, while Cert-$R^4$ excels

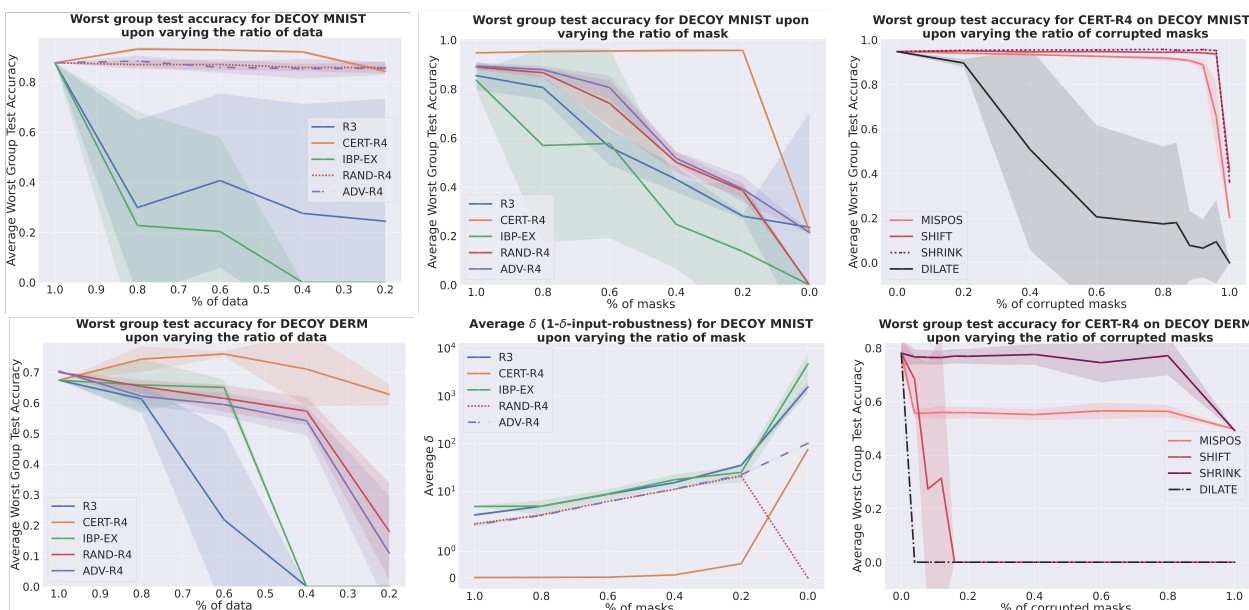

Figure 1: Experimental ablations to understand the performance of $R^4$. **Left column:** We plot the worst-group accuracy as we reduce the percentage of data and masks available at training time for DecoyMNIST (top) and DecoyDERM (bottom) **Center column:** We plot the worst-group accuracy (top) and average $\kappa$ (i.e. explanation fragility) as we reduce the percentage of masks available at training time for DecoyMNIST. **Right column:** We plot the $R^4$ worst-group accuracy as we vary the percentage of masks considering different types of corruptions for DecoyMNIST (top) and DecoyDERM (bottom).

in preserving performance, losing less than 10% accuracy in both datasets with the reduction of available information.

**Resilience to mask variations.** In the center column, we fix the dataset size and vary only the proportion of masks available during training. For each mask fraction, we maintain a constant MLX regularization strength (defined as the gradient magnitude for gradient-based methods or the adversarial loss for robustness-based methods) and scale the weight decay coefficient proportionally to the mask percentage. This setup isolates the effect of MLX regularization on sample complexity while minimizing the influence of weight regularization; we defer an in-depth analysis of the effects of weight decay to Appendix D. We find that both statistical and first-order variants of $R^4$ significantly outperform prior methods such as $R^3$ and IBP-Ex, which degrade rapidly as mask availability decreases. Notably, Cert-$R^4$ remains unaffected by missing masks (except at 0%) and achieves superior worst-group performance despite being trained without any weight decay, highlighting its sample efficiency. Finally, the low variance across runs in both center and left column plots further demonstrates the stability of our approach.

The center bottom plot shows a similar trend to the top plot, this time for explanation fragility. Although IBP-Ex and $R^3$ exhibit sharp increases in fragility as mask availability decreases, Cert-$R^4$'s $\kappa$-input-robustness remains stable until mask ratios drop significantly. This robustness stems from directly optimizing the bounds of input gradients. Consequently, $R^4$ not only avoids reliance on masked features but also remains resilient to distributional shifts in those features, even when mask coverage is limited. This is further supported by results on datasets such as ISIC (Table 1), where Cert-$R^4$ performs well despite masks being available for less than half of the samples in the dataset.

**Robustness of $R^4$ to mask corruptions.** We also examine the impact of corrupted masks, which can arise from human error or automated labeling inaccuracies. We consider four relevant corruption types: (i) 'misposition', where the mask is placed far from the non-core region; (ii) 'shift', involving slight displacement with partial overlap; (iii) 'shrink', and (iv) 'dilation', which reduce or expand the mask area without altering its position. Among these, 'dilation' is the most detrimental, which is unsurprising given that it also downweights

the contribution of core features. Shift and 'misposition' also degrade performance, particularly in datasets with large masks such as DecoyDERM, while 'shrink' has a minimal effect on $R^4$. These results suggest that 'dilation' and 'shift' corruptions are particularly harmful because they simultaneously suppress core features, which should remain informative, and reinforce non-core ones in regions that no longer align with the original, accurate masks. 'Misposition' errors, by contrast, are partially offset by high gradient activity in the misplaced regions. Therefore, interestingly, our ablation study suggests that practitioners should focus on mask quality rather than mask quantity when employing MLX approaches.

## 6 Discussion

We introduced a novel gradient- and robustness-based method to mitigate shortcut learning. We find that the strength of our proposed learning objective and its ability to lend itself well to a number of different approximations, achieves superior results across all datasets. Notably, the method is particularly effective in practical scenarios involving noisy annotations, highly complex models, and small, imbalanced datasets. Promising directions for future work include developing a theoretical framework to understand the role of weight regularization in these scenarios, leveraging techniques with statistical guarantees (e.g., randomized smoothing), and improving bound propagation tightness using methods such as CROWN. In the long term, advancing shortcut learning mitigation in large language models will require scalable methods that account for both semantic understanding, as well as the structure encoded in learned representations.

**Limitations.** Our method is primarily limited in settings where large models are used in combination with few or sparse masks and constrained computational resources. In such cases, one may need to employ Adv-RF or Rand-RF with reduced iterations or samples, yielding a less precise estimate of the perturbed maximum gradient in spurious regions. Another common drawback of techniques employing saliency maps, as noted by Li et al. (2023), is that when multiple shortcuts are present, reducing reliance on only some can lead models to overcompensate by relying more heavily on others. While this is a valid concern, we argue that it arises primarily during identification of non-core/shortcut features or mask-acquisition. This highlights the importance of developing methods with low failure rates in detecting such features and constructing accurate masks. However, provided that this condition is met, we believe that our approach can effectively and efficiently mitigate shortcut learning in these types of settings.

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

## A  Benchmark Datasets

**Decoy MNIST.** The first synthetic dataset we consider is Decoy-MNIST a popular MLX benchmark proposed in Ross et al. (2017). This a modification of the classical MNIST dataset which is comprised of 28 by 28 pixel images each representing a digit 0 through 9. To test shortcut learning, a small patch is added into the corners (with corners being selected randomly) of each image. The grey-scale color of the patch added to the training samples is exactly correlated with the label of the image while at test time the color of the patch has no correlation with the true label.

**Decoy DERM.** The second synthetic benchmark is a novel modification of the skin cancer classification "DermMNIST" benchmark Yang et al. (2023) that we term DecoyDERM. As in DecoyMNIST, DecoyDERM adds a small colored patch into the corner or each image, with the color of the patch being correlated with the label for training images and uncorrelated with the label for test images. Advantage of this benchmark compared to DecoyMNIST is (1) it mirrors a real-world, safety-critical scenario where shortcut learning has been observed and has potential adverse affects (2) is comprised of full-color medical images at a variety of resolutions from 28 by 28 up to 224 by 224. Thus this dataset is a strong benchmark for understanding the scalability and potential real-world effectiveness of MLX approaches.

**ISIC.** The ISIC dataset Codella et al. (2019) contains a number of benign and malignant skin lesions, which induce a skin cancer detection task for our MLX methods. Previous work that addressed *shortcut learning* Rieger et al. (2020) showed that because approximately 50% of the benign examples contain colourful patches, unpenalized DNNs rely on such spurious visual artifacts when attempting to detect sking cancer, thus generalizing poorly at inference time. As such, we follow the setup described in Rieger et al. (2020) to split the dataset into three groups: non-cancerous images without patch (NCNP) and with patch (NCP), as well as cancerous images (C), out of which only the NCP group contains associated masks.

**Plant Phenotyping.** The plant phenotyping dataset of "plant" for short is a dataset of microscopy images of plant leaves each representing a different phenotype that the model must classify. Previous studies have revealed that machine learning models tend to rely on the augur or solution that the plant sample is resting in rather than on phenotypic traits of the plant itself. Automatic feature extraction has been used identify and mask regions of each image constitute background (spurious) and foreground (non-spurious).

**Salient ImageNet.** The Salient ImageNet dataset (Singla et al., 2022) provides a series of human identified spurious correlations in the popular ImageNet dataset. Each mask determines if pixels belong to a "core" (non-shortcut) or "spurious" (shortcut) feature group. We use the 5000 available image and masks for this datset to understand the effectiveness of our approach when fine-tuning models (e.g., ResNet-18) rather than training from scratch.

**Decoy IMDB.** The IMDB dataset (Maas et al., 2011) is made up of 50000 IMDB movie text reviews, half of which (25000) are in the train set and the other half (25000) in the test set, and induces a binary classification sentiment analysis task. Previous works such as Wang & Culotta (2020) have identified and automatically extracted spurious features from this dataset, namely words that are correlated with a specific class. An example is the word "*spielberg*", which is the name of a well-known movie director that is correlated with a positive sentiment, due to the largely successful and appealing to audience movies he has produced. We make this task harder by prepending to every train set positive-sentiment review the word "*spielberg*" and at the beginning of every train set negative-sentiment review the word "*jonah*". For every example of the test set, one of the two words above is chosen at random and inserted at a random position in the sentence. We employ the method of Wang & Culotta (2020) to extract spurious words.

## B  Model Architecture

## Decoy MNIST.

```
Sequential(
    (0): Linear(in_features=784, out_features=512, bias=True)
    (1): ReLU()
```

```
    (2): Linear(in_features=512, out_features=10, bias=True)
)
```

## Decoy DERM.

```
Sequential(
    (0): Conv2d(3, 32, kernel_size=(3,3), stride=(1,1), padding=(1,1))
    (1): ReLU()
    (2): Conv2d(32, 32, kernel_size=(4,4), stride=(2,2), padding=(1,1))
    (3): ReLU()
    (4): Conv2d(32, 64, kernel_size=(4,4), stride=(1,1), padding=(1,1))
    (5): ReLU()
    (6): Conv2d(64, 64, kernel_size=(4,4), stride=(2,2), padding=(1,1)),
    (7): torch.nn.ReLU()
    (8): Flatten(start_dim=1, end_dim=-1)
    (9): Linear(in_features=14400, out_features=1024, bias=True)
    (10): ReLU()
    (11): Linear(in_features=1024, out_features=1024, bias=True)
    (12): ReLU()
    (13): Linear(in_features=1024, out_features=2, bias=True)
)
```

## ISIC.

```
Sequential(
    (0): Conv2d(3, 16, kernel_size=(4,4), stride=(2,2), padding=(0,0))
    (1): ReLU()
    (2): Conv2d(16, 32, kernel_size=(4,4), stride=(4,4), padding=(0,0))
    (3): ReLU()
    (4): Conv2d(32, 64, kernel_size=(4,4), stride=(1,1), padding=(1,1))
    (5): ReLU()
    (6): Flatten(start_dim=1, end_dim=-1)
    (7): Linear(in_features=43808, out_features=100, bias=True)
    (8): ReLU()
    (9): Linear(in_features=100, out_features=2, bias=True)
)
```

## Plant Phenotyping.

```
Sequential(
    (0): Conv2d(3, 32, kernel_size=(3,3), stride=(1,1), padding=(1,1))
    (1): ReLU()
    (2): Conv2d(32, 32, kernel_size=(4,4), stride=(2,2), padding=(1,1))
    (3): ReLU()
    (4): Conv2d(32, 64, kernel_size=(4,4), stride=(1,1), padding=(1,1))
    (5): ReLU()
    (6): Conv2d(64, 64, kernel_size=(4,4), stride=(2,2), padding=(1,1)),
    (7): torch.nn.ReLU()
    (8): Flatten(start_dim=1, end_dim=-1)
    (9): Linear(in_features=173056, out_features=1024, bias=True)
    (10): ReLU()
    (11): Linear(in_features=1024, out_features=1024, bias=True)
    (12): ReLU()
    (13): Linear(in_features=1024, out_features=2, bias=True)
)
```

## Salient ImageNet.

As the pre-trained backbone model, we use **ResNet-18** (He et al., 2016), to which we attach the following classification head:

```
Sequential(
    (0): Identity()
    (1): Dropout(p=0.5)
    (2): Linear(in_features=512, out_features=6, bias=True)
)
```

## Salient ImageNet.

As the pre-trained backbone model, we use **BERT** (Devlin et al., 2019), to which we attach the following classification head:

```
Sequential(
    (0): Linear(in_features=768, out_features=384, bias=True)
    (1): ReLU()
    (2): Linear(in_features=384, out_features=2, bias=True)
)
```

## C  Extension of Gradient-Based Regularization Techniques to Language Modeling Tasks

$\mathbf{R}^3$. This extension requires minimal modifications to the standard training pipeline. First, we identify the positions of spurious tokens within the input sequence. For each of these tokens, we extract their embedding, which includes both token and positional components. Lastly, we compute the gradient of the model's loss with respect to these embeddings and augmenting the original loss function with the norm of the computed gradients.

**Smooth-$\mathbf{R}^3$** and **Rand-$\mathbf{R}^4$**. In contrast to approaches that use continuous normal noise (e.g., Gaussian perturbations applied to image pixels), we propose a discrete perturbation mechanism suited to language. Specifically, we define a hyperparameter $\alpha \in [0, 100]$ that controls the percentage of token substitutions applied to a given input text.

In Smooth-$\mathrm{R}^3$, $\alpha\%$ tokens are randomly sampled from the entire input text and replaced with alternatives drawn uniformly at random from the full vocabulary. In Rand-$\mathrm{R}^4$, replacements are restricted to a predefined set of spurious words (for example, the ones identified in the IMDB dataset (Maas et al., 2011) by Wang & Culotta (2020)), and substitutions are applied only to tokens in the input that match this spurious set.

This randomized substitution process is repeated $n$ times per input example, where $n$ is a user-defined sampling hyperparameter. For each perturbed sample, we compute the gradient of the loss function with respect to the token embeddings. Finally, the regularization term added to the original loss function is the mean of the gradients across the $n$ samples in the case of Smooth-$\mathrm{R}^3$, and the element-wise maximum of the gradients across samples for Rand-$\mathrm{R}^4$, respectively.

**Adv-$\mathbf{R}^4$**. To enhance the adversarial regularization capabilities of Adv-RF, we draw methodological inspiration from the $GCG$ attack proposed by Zou et al. (2023). This extension leverages gradient-based token substitution in an iterative manner to construct adversarial variants of input sequences.

The process begins by identifying an ordered sequence of spurious tokens present in the original input. For each token in this sequence, we perform a search over a predefined vocabulary of spurious words to identify the candidate replacement that maximizes the gradient of the model's loss with respect to the embedding of the current token. Formally, for a given spurious token $w_i$ in the input, we select a replacement $w_i' \in \mathcal{S}$ such that:

$$w_i' = \arg \max_{w \in \mathcal{S}} \|\nabla_{e(w)} \mathcal{L}(x^{(i \to w)})\|$$

where $\mathcal{S}$ is the set of known spurious words, $e(w)$ is the embedding of token $w$, and $x^{(i \rightarrow w)}$ denotes the input sequence with the $i$-th token replaced by $w$.

After each substitution, the modified input sequence is updated, and the process is repeated for the next spurious token in the sequence. This greedy, iterative attack continues until all spurious tokens have been considered and replaced. The resulting adversarially perturbed sequence is then used to compute the regularization term added to the original loss function.

## D    Impact of Weight Decay

As discussed in Section 5, we opted to set the weight decay coefficient to be proportional to the ratio of masks when performing mask sample complexity ablations. However, prior to this experimental design decision, we set out to perform a vanilla ablation, leaving all the training parameters, including the ones pertaining to weight regularization, the same as in the case of the best performing model achieving the results show in Table 1. However, this yielded surprising results, namely that all methods which used weight regularization varied insignificantly in worst (and macro average) group accuracy, regardless of the mask percentage available at training time. This, naturally, led us to hypothesize that in numerous previous MLX methods, either the weight regularization is the parameter that actually controls the robustness to spurious features the most, or, at the very least, it is indispensable in achieving previous literature results and *must* be used in addition to the MLX objective.

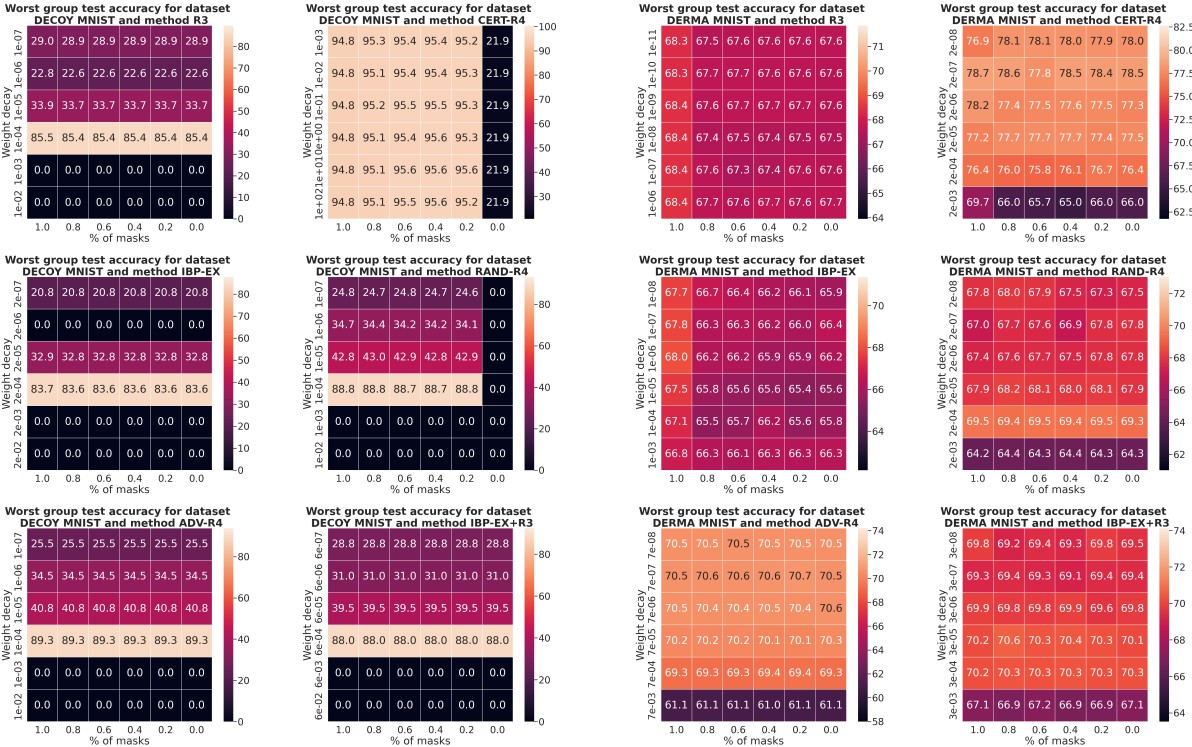

Figure 2: Sample complexity ablation varying the percentage of masks available at train time for different amounts of weight regularization. We show the results for DecoyMNIST in the two leftmost columns and the results for DecoyDERM in the rightmost two columns.

To explore the connection between spurious feature reliance and weight regularization, we perform the same ablations as in Section §5.5 for different values of weight decay, and then plot the results as a heat map for each MLX technique, as shown in Figure 2. We first note that the initial weight decay parameter used corresponds to the y-axis label of the fourth row, for every heatmap. Starting with the results on DecoyMNIST (first and second columns starting from the left), we observe that for non-$R^4$ methods, the performance of the algorithm quickly decreases as weight decay strays away from its initial value, where the best performance, or "mode" is achieved. Adv-$R^4$'s behaviour is similar, but the effect is slightly weaker. Notably, the accuracy of all these four methods does not vary *at all* along the x-axis, suggesting that independence from spurious features is highly correlated with weight decay, but less with the mask ratio. Lastly, Cert-$R^4$ (second top-most plot), which we highlight is trained on this dataset for performance **without any weight regularization** has little variation in group accuracy regardless of the $\ell_2$ regularization magnitude due to the strength of the MLX term used in training. The only parameter space location where its accuracy collapses is when we do not have access to any masks whatsoever, which uncovers an interesting behaviour: weight decay does not influence it at all, since the performance in the middle row (weight decay 0) is the same as any other row.

The heat maps for DecoyDERM are slightly harder to interpret, because it is (i) a much more complex dataset and (b) a harder MLX objective (regions taken up by swatches are larger). For the majority of methods, the ablations performed on this dataset do indeed show a decrease, albeit slight, along the x axis (i.e. varying the mask ratio), showing that in a real-world scenario, as complexity of the task increases, so does the importance of high-quality masks. Turning back to the impact of weight decay, we can see that especially in methods such as $R^3$ or IBP-Ex+$R^3$ it plays a huge role, noticeable from the little variation across both x and y axis, whereas in, for example, Cert-$R^4$ (top-right corner), the accuracy seems to be varying more. Lastly, one thing that can equally be learned equally from all algorithms used to ensure robustness to spurious features in DecoyDERM is that for complex datasets weight decay is, next to the strength given to MLX regularization, one of the most critical parameters. As such, practicians must make sure not to over, nor under-regularize with respect to weights, because the large number of model parameters in certain architectures (especially large convolutional ones) might determine the weight decay term to far exceed the MLX one and thus determine the optimizer to ignore the latter, or viceversa, might make the objective impossible to optimize due to the difficulty of the MLX term.

# E   Model Size Ablations

We also perform ablations of model architecture sizes, by varying the number of layers at train time for DecoyMNIST and DecoyDERM, as can be seen in Figure 3. For the first, the performance results obtained in Table 1 are achieved by using a 1 layer fully-connected network with 512 hidden units (corresponding to the '1-layer' x-axis label), and we vary the number of layers, while maintaining the amount of hidden units in each layer, which is to say we ablate the depth. For the latter, the performance results are obtained with the 'Medium-Large' architecture, consisting of 4 convolutional layers and 3 fully-connected layers, and we perform the ablation by adding or removing 1 convolutional layer and 1 fully connected layer per x axis step.

Looking at the top row, depicting the worst group accuracy per model architecture for different MLX techniques in both datasets, we can observe that the statistical and first-order adversarial $R^4$ approaches yield improved generalization when compared to IBP-Ex and $R^3$ for different architectures, being in some cases 40% more worst group accurate (for example at the '3 layers' x label value). Reassuringly, for complex architectures as is the case with DecoyDERM, both the variations between subsequent architectures and their respective standard deviation (measured across runs) stay low, property which might be desirable for end users when deployed in critical systems. One interesting behaviour emerges, though, in Cert-$R^4$: it achieves significantly higher accuracy by using the initial performance-optimized architecture. When varied, the accuracy drops rapidly, suggesting that Cert-$R^4$ requires a very specific set of parameters to function as expected, and thus might need re-optimizing when a certain parameter changes. This effect is compounded with the usage of convex relaxation techniques (i.e. IBP), which are know to generalize poorly to large architectures, achieving vacuous bounds, and the complexity of the task, which requires a large number of parameters in order to be expressive. However, lastly, a positive behaviour we can notice is the fact that the explanation fragility remains constantly very close to 0, meaning that even if we might be independent to a subset of spurious features, we are *very sure* this particular algorithms will never be fooled by noise added

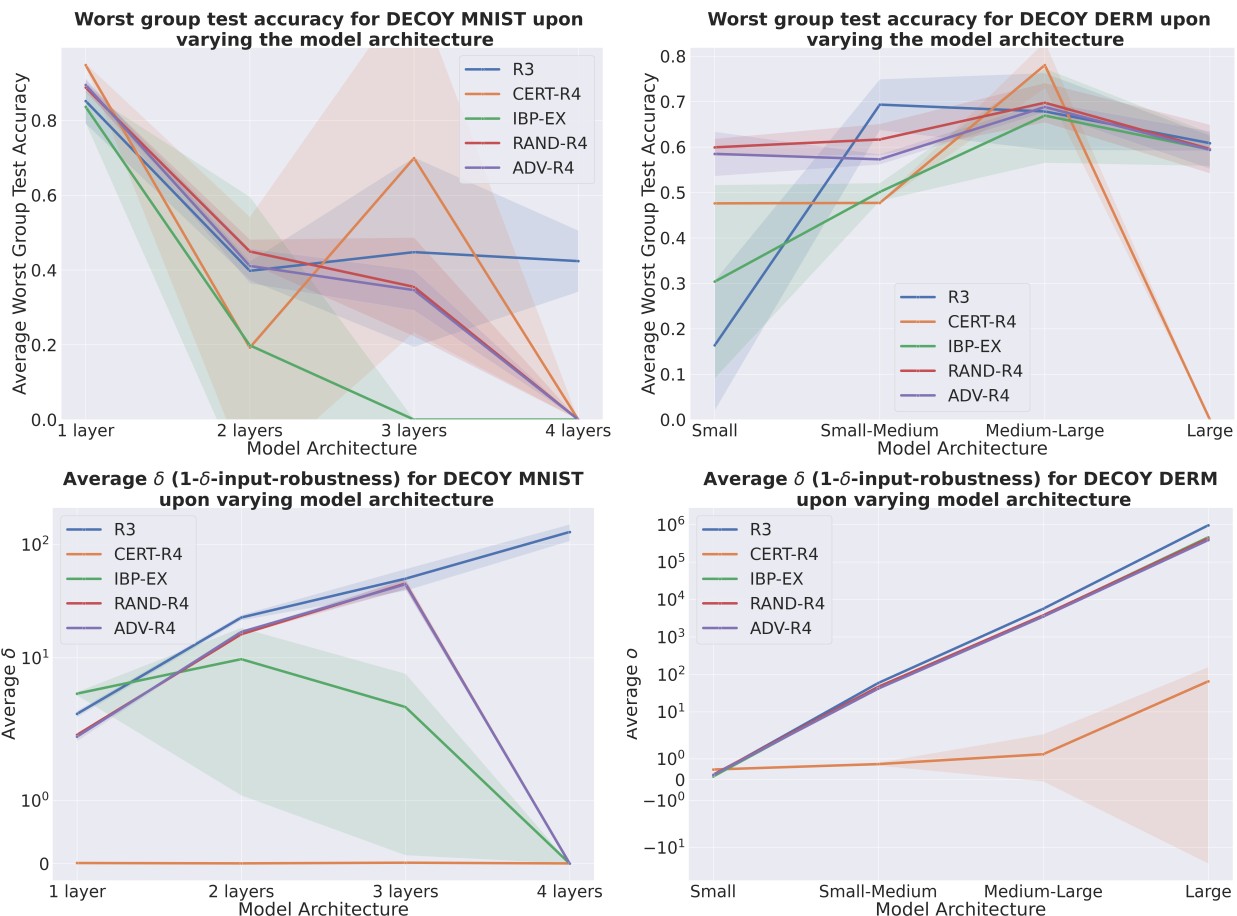

Figure 3: Model size ablation varying the number of layers in the model architecture at train time. We plot worst case group accuracy (top) and explanation fragility (bottom) for DecoyMNIST (**left column**) and DecoyDERM (**right column**).

to those features. This leads us to think that there is a trade-off between explanation fragility and group accuracy (in Cert-R$^4$), and coming up with a way of dynamically loosening the $\delta$-input-robustness guarantees during training might yield improved worst-group accuracy.

## F    Additional Mask Corruption Ablations

In Figure 4 we reuse the mask corruption ablation experimental setup of Figure 1 and plot the worst group accuracy and gradient fragility ($\delta$) when training a model on DecoyMNIST using R$^3$and, respectively, IBP-Ex. Firstly, we can immediately notice that as the corruption ratio at train time increases, each method's performance decreases at almost exactly the same rate across the corruption type applied, a plausible explanation for these dynamics being the reliance of the methods on weight regularization (as was demonstrated in Appendix D). Being corruption-agnostic suggests that previous MLX techniques are less expressive and flexible compared R$^4$, failing to exploit fine-grained spatial properties of annotations and having a coarse train-induced mechanism to mitigating shortcut learning. Secondly, both R$^3$and IBP-Ex perform worse than Cert-R$^4$when the mask corruption applied is either 'shrink', 'shift' or 'misposition', and slightly better when it is 'dilation', behaviour that can be explained by the strong regularization magnitude of R$^4$induced by the maximization term in  equation 3. Lastly, we observe that as the ratio of corrupted annotations increases, the gradient fragility of both techniques has a large enough magnitude such that we

can consider that for all practical purposes, $R^3$ and IBP-Ex have zero robustness to perturbations of the shortcut features, like those introduced by Dombrowski et al. (2019).

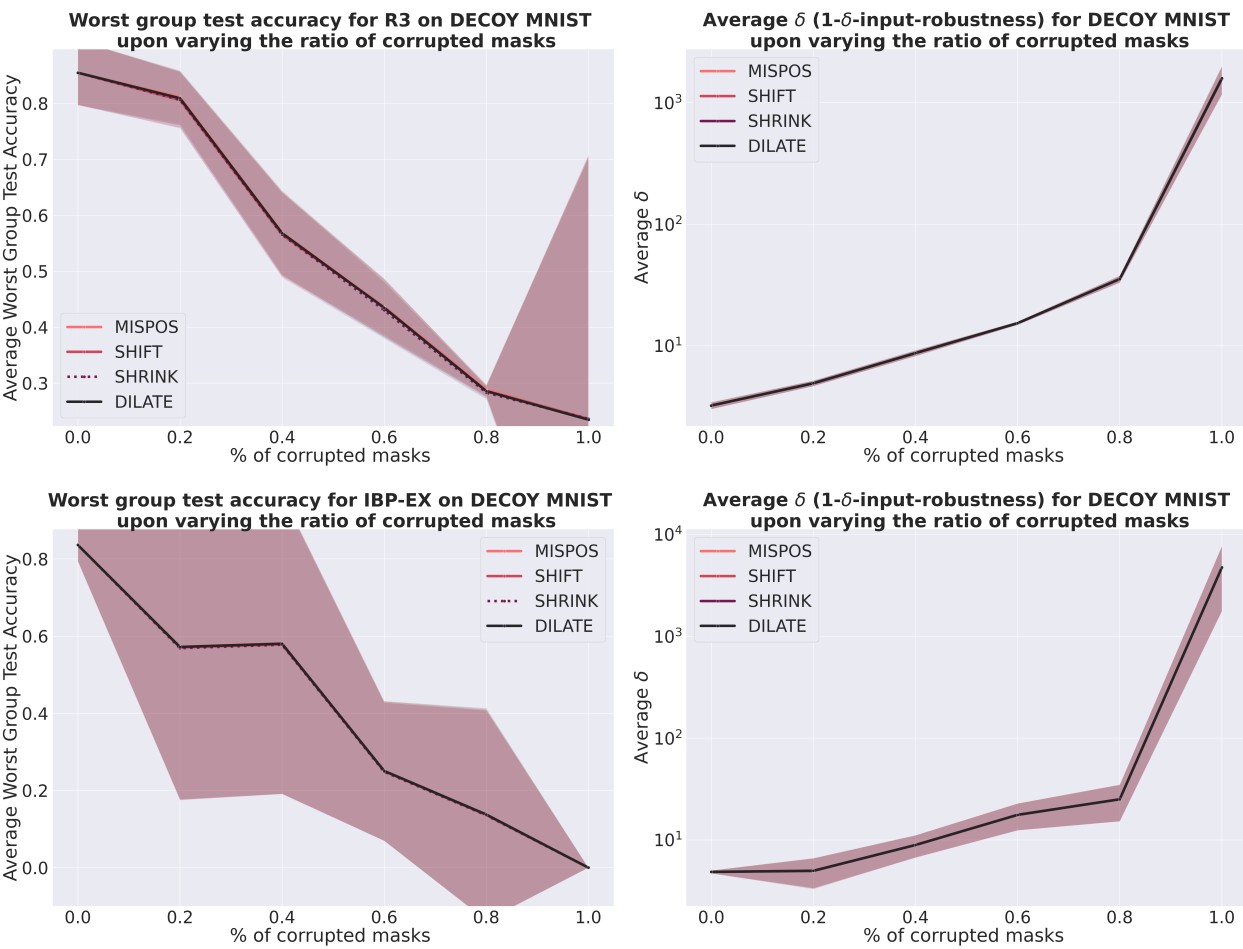

Figure 4: Mask corruption ablations using the experimental setup in Figure 1. We plot worst case group accuracy (left column) and explanation fragility (right column) for DecoyMNIST using $R^3$ (top row) and IBP-Ex (bottom row).

# G Additional Experimental Details

Table 3: Robustness to Perturbation of Spurious Features

| | Learning Objective | | | | | | | |
|---|---|---|---|---|---|---|---|---|
| | ERM | $R^3$ | Smooth-$R^3$ | IBP-Ex | IBP-Ex + $R^3$ | Rand-$R^4$ | Adv-$R^4$ | Cert-$R^4$ |
| **Dataset ↓** | **Avg $\delta$** | **Avg $\delta$** | **Avg $\delta$** | **Avg $\delta$** | **Avg $\delta$** | **Avg $\delta$** | **Avg $\delta$** | **Avg $\delta$** |
| DecoyMNIST | 497.44 | 4.35 | 4.33 | 5.61 | 3.43 | 2.10 | 2.02 | **0.93** |
| DecoyDERM | 6742.41 | 5762.81 | 3906.13 | 2979.48 | 2892.32 | 3790.20 | 3504.75 | **1.297** |
| ISIC | 0.423 | **0.00167** | 1.029 | 1.784 | 2.878 | 1.392 | 0.400 | 0.901 |
| Plant | 4011.40 | 4519.85 | 2082.06 | 4742.33 | 1105.28 | 1285.71 | 1787.99 | **89.38** |

Table 4: Robustness to Perturbation of Core Features

| | Learning Objective | | | | | | | |
|---|---|---|---|---|---|---|---|---|
| | ERM | $R^3$ | Smooth-$R^3$ | IBP-Ex | IBP-Ex + $R^3$ | Rand-$R^4$ | Adv-$R^4$ | Cert-$R^4$ |
| Dataset $\downarrow$ | Avg $\delta$ | Avg $\delta$ | Avg $\delta$ | Avg $\delta$ | Avg $\delta$ | Avg $\delta$ | Avg $\delta$ | Avg $\delta$ |
| DecoyMNIST | 332.3 | 7.7 | 7.6 | 8.7 | 6.4 | 6.2 | 6.3 | 39.4 |
| DecoyDERM | 7918.7 | 5811.9 | 3925.1 | 3714.5 | 3684.6 | 3800.2 | 3558.9 | 453.7 |
| ISIC | 0.45 | 0.002 | 1.17 | 1.92 | 3.06 | 1.6 | 0.45 | 1.07 |
| Plant | 4410.9 | 4657.3 | 2034.7 | 4881.83 | 1223.02 | 1307.2 | 2439.8 | 97.8 |

# H   Model Outputs Upper Bound for Adversarially Perturbed Spurious Regions - Proof

We firstly restate our claim and then provide a proof.

*Proposition* 1. Given a function $f$ induced by a model parametrized by $\theta$, trained on dataset $\mathcal{D} = \{(\boldsymbol{x}^{(i)}, \boldsymbol{y}^{(i)}, \boldsymbol{m}^{(i)})\}_{i=1}^N$ with loss function $R^4$ as in equation 3, adversarial perturbations of magnitude at most $\epsilon$ (i.e. with $x' \in \mathcal{B}_\epsilon^{\boldsymbol{m}}(x)$) and post-hoc computed $\delta$-input robustness of magnitude at most $\delta^*$, then the output difference norm between any perturbed input and its standard counterpart is upper bounded by:

$$\|f^{\boldsymbol{\theta}}(\boldsymbol{x}) - f^{\boldsymbol{\theta}}(\boldsymbol{x}')\| \leq \delta^\star \|\epsilon\|(1 + \frac{1}{2}\|\epsilon\|).$$

*Proof.* We begin by writing the first-order Taylor approximation, with second-order remainder term at the spurious (masked) region of input $x'$, i.e. $\mathcal{B}_\epsilon^m(\boldsymbol{x}')$. In that case, the function $f_{\boldsymbol{m},2}^{\boldsymbol{\theta}}(\boldsymbol{x})$ is *exactly equal to*:

$$f_m^{\boldsymbol{\theta}}(\boldsymbol{x}) = f^{\boldsymbol{\theta}}(\boldsymbol{x}') + \underbrace{\nabla_x f^{\boldsymbol{\theta}}(\boldsymbol{x}')^\top (\boldsymbol{m} \odot (\boldsymbol{x} - \boldsymbol{x}'))}_{\text{function change}} + \underbrace{\mathcal{R}_2(f_m^{\boldsymbol{\theta}}(\boldsymbol{x}))}_{\text{second-order remainder}}$$

By rearranging and applying the norm on both sides, we have:

$$\|f_m^{\boldsymbol{\theta}}(\boldsymbol{x}) - f^{\boldsymbol{\theta}}(\boldsymbol{x}')\| = \|\nabla_x f^{\boldsymbol{\theta}}(\boldsymbol{x}')^\top (\boldsymbol{m} \odot (\boldsymbol{x} - \boldsymbol{x}')) + \mathcal{R}_2(f_m^{\boldsymbol{\theta}}(\boldsymbol{x}'))\|$$

We note that, as mentioned in §4.1, by optimizing $R^4$'s objective, we can bound the gradient magnitude for any point in the $\epsilon$-ball of the masked region: $\forall \boldsymbol{x}^\star \in \mathcal{B}_\epsilon^m(\boldsymbol{x}'), \|\nabla_x f^{\boldsymbol{\theta}}(\boldsymbol{x}^\star)\| \leq \delta^\star$. Using the property given by our technique, as well as the triangle inequality, we have that:

$$\|f^{\boldsymbol{\theta}}(\boldsymbol{x}) - f^{\boldsymbol{\theta}}(\boldsymbol{x}')\| \leq \delta^\star \|\boldsymbol{m} \odot (\boldsymbol{x} - \boldsymbol{x}')\| + \|\mathcal{R}_2(f_m^{\boldsymbol{\theta}}(\boldsymbol{x}'))\|$$

Firstly, we remind the reader of *Taylor's Theorem with Lagrange Remainder*. We write the $n$-th order Taylor's approximation for a function $f$ at a point $x' \in [x, x+h]$ and assume that $f$ is $n+1$ differentiable in this interval. The theorem tells us that the norm of the remainder term of order $n+1$ is upper bounded by:

$$\|\mathcal{R}_{n+1}(f(\boldsymbol{x}))\| \leq \frac{M}{(n+1)!}\|h\|, \text{ where } M = sup_{x' \in [x, x+h]}(\nabla_x^{n+1} f(\boldsymbol{x}))$$

In our case, we have by definition that $x' \in \mathcal{B}_\epsilon^m(\boldsymbol{x})$, which implies $\|\boldsymbol{x} - \boldsymbol{x}'\| \leq \|\boldsymbol{m} \odot \epsilon\| \leq \|\epsilon\|$, since $m \in [0, 1]$. We also note that $\|h\| = \|\boldsymbol{m} \odot (\boldsymbol{x} - \boldsymbol{x}')\|$ in the above equation. Lastly, as mentioned in §4.1, because the norm of the gradient in the masked $\epsilon$-ball is bounded by $\delta^*$, then trivially the norm of the Hessian in the same region is also bounded by $\delta^*$ and corresponds with the notion of $M$ defined above, for a first-order Taylor-approximation with second-order remainder. Lastly, since the matrix norm is submultiplicative, we obtain the following **strict upper bound**:

$$\|f^{\boldsymbol{\theta}}(\boldsymbol{x}) - f^{\boldsymbol{\theta}}(\boldsymbol{x}')\| \leq \delta^\star \|\epsilon\|(1 + \frac{1}{2}\|\epsilon\|).$$

$\square$

# I  Why is R$^4$ better than IBP-Ex?

The following result demonstrates the merit of R$^4$ over IBP-Ex.

*Proposition* 2. Given a dataset $\mathcal{D} = \{\boldsymbol{x}^{(i)}, y^{(i)}\}_{i=1}^{n}$, we wish to fit a linear regressor that predicts the output well without using the pre-specified $k$ irrelevant features (of the total $l$ features), which we assume are the last k dimensions of $\boldsymbol{x}$ without loss of generality. We denote by the weights fitted using IBP-Ex and R$^4$ by $\boldsymbol{w}_I$ and $\boldsymbol{w}_R$ respectively. The following result holds for the norm of irrelevant features.

$$\|\boldsymbol{w}_I \odot \boldsymbol{m}\| \leq \sqrt{k}\tau/\|\epsilon\|, \quad \|\boldsymbol{w}_R \odot \boldsymbol{m}\| \leq \tau$$

$$\text{where } \boldsymbol{m} = [\overbrace{0, 0, \ldots, \underbrace{1, 1, \ldots}_{k}}^{l}] \text{ and } \tau \triangleq \max_{\boldsymbol{x} \in \mathcal{D}, \boldsymbol{x}' \in B^{\epsilon}(\boldsymbol{x})} |\boldsymbol{w}_*^T \boldsymbol{x}' - \boldsymbol{w}_*^T \boldsymbol{x}|$$

We require that the norm of the irrelevant features to be as small as possible. However, the bound on the norm of irrelevant features given by IBP-Ex is much weaker than that of R$^4$. Because the upper bound given by IBP-Ex gets weaker with the number of irrelevant features and is further exacerbated by the practically small value of $\epsilon$. On the other hand, R$^4$ fitted $\boldsymbol{w}_R$ bounds the norm of irrelevant features well even when their dimension is high. Intuitively, the error in suppressing the function deviation of the inner maximization loop blows up with the number of dimensions in IBP-Ex. On the other hand, since R$^4$ suppresses the norm directly, the support of irrelevant features is kept in check.

*Proof.* We will generalize the definition of $\tau$ to work for both IBP-Ex or R$^4$ by generalizing the function $\boldsymbol{w}^T\boldsymbol{x}$ to $f(\boldsymbol{x})$. The more general definition of $\tau$ is as follows.

$$\tau \triangleq \max_{\boldsymbol{x} \in \mathcal{D}, \boldsymbol{x}' \in B^{\epsilon}(\boldsymbol{x})} |f(\boldsymbol{x}') - f(\boldsymbol{x})|$$

We first prove the result for $\boldsymbol{w}_I$, which bounds the function value deviation in the $\epsilon$-neighborhood.

Since we are fitting a linear regressor, the coefficients of the $i^{th}$ irrelevant feature
$w_i$ is $\{f(\boldsymbol{x} + [0, \ldots, 0, \underbrace{\|\epsilon\|}_{i^{th}}, 0, \ldots, 0]) - f(\boldsymbol{x})\}/\|\epsilon\| \leq \tau/\|\epsilon\|$. Therefore, the norm of the $k$ irrelevant features is bounded by $\sqrt{k}\tau/\|\epsilon\|$.

The result for $\boldsymbol{w}_R$ follows directly from observing that R$^4$ minimizes the gradient norm of irrelevant features, which is $\boldsymbol{w}_R \odot \boldsymbol{m}$, and from the generalized definition of $\tau$. $\qquad\square$

# J  Interval Arithmetic for Bounding Input Gradients

We begin by recalling the form of the forward and backwards pass sated in the main text where we have a neural network model $f^{\boldsymbol{\theta}} : \mathbb{R}^{n_{\text{in}}} \to \mathbb{R}^{n_{\text{out}}}$ with $K$ layers and parameters $\boldsymbol{\theta} = \{(W^{(i)}, b^{(i)})\}_{i=1}^{K}$ as:

$$\hat{z}^{(k)} = W^{(k)} z^{(k-1)} + b^{(k)},$$
$$z^{(k)} = \sigma\left(\hat{z}^{(k)}\right)$$

where $z^{(0)} = x$, $f^{\boldsymbol{\theta}}(\boldsymbol{x}) = \hat{z}^{(K)}$, and $\sigma$ is the activation function, which we assume is monotonic. We have the backwards pass starting with $\delta^{(L)} = \nabla_{\hat{z}^{(L)}} f^{\boldsymbol{\theta}}(\boldsymbol{x})$, we have that backwards pass is given by:

$$\delta^{(k-1)} = \left(W^{(k)}\right)^{\top} \delta^{(k)} \odot \sigma'\left(\hat{z}^{(k-1)}\right)$$

Where we are interested in $\delta^{(0)} = \nabla_x f^{\boldsymbol{\theta}}(\boldsymbol{x})$.

As highlighted in the paper the above forwards and backwards bounds consist solely of matrix multiplication, addition, and the application of a non-linearity.

Where we start with an input interval $[x^L, x^U]$, we must define interval versions of the above operations such that they maintain that the output interval of our interval operations provably contains all possible outputs of their non-interval counterparts. We will represent interval matrices with bold symbols i.e., $\boldsymbol{A} := [A_L, A_U] \subset \mathbb{R}^{n_1 \times n_2}$. We denote interval vectors as $\boldsymbol{a} := [a_L, a_U]$ with analogous operations.

**Definition J.1** (Interval Matrix Arithmetic). Let $\boldsymbol{A} = [A_L, A_U]$ and $\boldsymbol{B} = [B_L, B_U]$ be intervals over matrices. Let $\oplus, \otimes, \odot$ represent interval matrix addition, matrix multiplication and element-wise multiplication, such that

$$
\begin{aligned}
A + B &\in [\boldsymbol{A} \oplus \boldsymbol{B}] \quad \forall A \in \boldsymbol{A}, B \in \boldsymbol{B}, \\
A \times B &\in [\boldsymbol{A} \otimes \boldsymbol{B}] \quad \forall A \in \boldsymbol{A}, B \in \boldsymbol{B}, \\
A \circ B &\in [\boldsymbol{A} \odot \boldsymbol{B}] \quad \forall A \in \boldsymbol{A}, B \in \boldsymbol{B}.
\end{aligned}
$$

Both the addition (defined element-wise) and element-wise multiplication of these bounds can be accomplished by simply taking all 4 combinations of the interval end points and returning the maximum and minimum. The matrix multiplication is slightly more complex but can be bounded using Rump's algorithm Rump (1999). These operations are standard interval arithmetic techniques and are computed in at most $4\times$ the cost of a standard forward and backward pass.

For the non-linearity, we have assumed the function is monotonic which is defined by: $x < y \implies \sigma(\boldsymbol{x}) \leq \sigma(y)$. Thus the element-wise application of $\sigma$ to an interval over vectors $[v^L, v^U]$ we simply have that the interval output is $[\sigma(v^L), \sigma(v^U)]$.

Taken together, these suffice to bound the output of a forward and backwards pass. For a more rigorous treatment we reference readers to (Wicker et al.).

