# OpenReview forum: "Model Guidance via Robust Feature Attribution"
_TMLR — Accepted by TMLR_

### Review · Reviewer_NY1T · 2025-07-08

**Summary Of Contributions:**

Modern neural network classifiers are highly accurate, especially when testing on data drawn from the same distribution as the training data.  However, these models are known to learn shortcuts in the data, such that the patterns that they learn to associate with each class label are not robust to distribution shifts or minority groups in the data.  To combat this, a variety of approaches to machine learning from explanations (MLX) have been proposed, where ground-truth feature annotations are used to guide the model towards learning more robust and interpretable solutions.  In particular, model explanations are used to depress the amount of attention paid to spurious features, so that the model can be “right for the right reasons”.  One shortcoming of this approach to MLX is that the salience maps produced by inspecting input gradients are known to be brittle, where small perturbations to the input cause large changes to the map.

The contribution of this work is to address this challenge, proposing R4, an extension of R3 with a focus on robustifying the gradient-based attributions.  As their initial formulation is intractable, the authors propose 3 methods for computing and optimizing their objective.  Results are offered across a range of tasks and in comparison to a range of baselines, showing R4 to be a strong choice of optimizer for MLX with feature salience.

**Audience:**

Yes

**Claims And Evidence:**

Yes

**Requested Changes:**

In the experiments section:
- Explain why Cert-R4 is not applied to Decoy IMDB and Salient Imagenet in Table 1.
- Explain why Cert-R4 performs (relatively) poorly on Plant in Table 2.
- Move some architecture ablation results and discussion to main paper.

Writing:
- End-to-end proofread and polish
- Edit introduction for clarity

(Suggested) Illustrative Figure 1

**Strengths And Weaknesses:**

## Strengths

This is a technically solid paper, and in my opinion merits acceptance (given the requested changes below).  In particular, the authors offer a well-motivated approach to a problem of high interest in the community.  Technical material is clearly presented, and experimental results are generally strong.  Code and other necessary information to reproduce and build on the experimental results are include.


## Weaknesses

I think the primary weakness of this paper is that the experimental results focus too much on showing the superiority of R4 to other methods.  This leaves a missed opportunity to analyze and discuss where and when R4 does not perform as strongly, which is important information to share with researchers aiming to build on your work.

For example, it would be interesting to have the authors discuss why Cert-R4 performs poorly on Plant in Table 2.  The current results discussion includes discussion of the other 3 datasets in Table 2, but does not mention this result.  Also, why is Cert-R4 not applied to Decoy IMDB and Salient Imagenet in Table 1?  Are there some situations where each of the 3 proposed variants are more or less appropriate, or possible cannot be applied?

Also in the same vein, I think the main paper should include some results and discussion regarding the ablations of model architectures, which are currently in the appendix, and not referenced in any detail in the main paper.  Appendix Figure 3 makes it seem like model architecture plays a significant role in determining the relative success of the various methods, so (at least some of) those results should be presented and discussed in the main paper.

My second high-level critique of the paper is that the writing would benefit from further careful editing and polish.  For example:
- In the introduction, “Learning such shortcuts poses a great challenge with deployment in safety-critical domains” is unclear (i.e., could be read that it’s hard to learn the shortcuts) and somewhat awkwardly phrased (“poses a great challenge with deployment”)
- In related works, “Mitigating shortcut learning has been approached from multiple fronts, the three major themes are…”
- In preliminaries, first sentence, the definition of a machine learning model actually seems to be the definition of a classification model.
- Section 4.1, the following needs a comma: “We begin by describing the theoretical intuition of our learning objective describing its benefits relative to prior works.”

Also, I found myself confused at the beginning of the paper as to exactly what types of “features” were being considered, before I finally realized that the paper focuses on gradient-based saliency maps.  For example, the term “masked” is introduced before it is specified that we are interested in gradient-based saliency maps, and thus it is unclear what it means at first and I had to re-read those paragraphs to fully understand.  The introduction should be edited for logical flow and clarity.

Finally, I would suggest that the paper could benefit from some sort of illustrative figure 1 orienting the reader towards the problem at hand (e.g., a 3 panel figure showing 1 typical saliency map, 1 illustrating brittleness to small perturbations, and 1 showing that r4 is robust to the same perturbation).

---

> ### Author Response · Authors · 2025-08-13
>
> We thank the reviewer for the consideration given to our work and address their concerns and comments below.
>
> 1. *“This leaves a missed opportunity to analyze and discuss where and when R4 does not perform as strongly [...] For example, it would be interesting to have the authors discuss why Cert-R4 performs poorly on Plant in Table 2.”* - The reviewer poses a valid point regarding the need of the research community to understand both the advantages and disadvantages of our novel methodology. To clarify, using their example, the reason why the model trained on Plant using Cert-R4 doesn’t yield an improvement margin as large as in the case of the other datasets is twofold. Firstly, Cert-R4 uses the Interval Bound Propagation method of Gowal et. al. (2018), which obtains worst-case predictions for perturbations inside a user-defined infinity-norm bounded ball, per pixel. Secondly, the saliency maps for Plant, which represent the nutrition solution the plant tissue is observed in, are larger compared to other datasets, oftentimes taking up more than half an image’s pixels. The fact that Cert-R4 induces the strongest regularizer among our techniques and is also prone to overapproximation, combined with the small number of core features compared to shortcut (masked) features hinders the model’s utility to some extent, which explains the result. In contrast, Cert-R4 performs best when trained on datasets that contain sparse(r) explanations (ISIC) or have saliency maps with a small area (DecoyIMDB). We have added a discussion of this at the end of Section 5.3 in our experimental section.
>
> 2. *“Also, why is Cert-R4 not applied to Decoy IMDB and Salient Imagenet in Table 1? Are there some situations where each of the 3 proposed variants are more or less appropriate, or possible cannot be applied?”* - The reviewer is right when saying that the three proposed variants are more or less appropriate to different use cases. Specifically, due to the fact that Cert-R4 uses Interval Bound Propagation, which suffers from overapproximation when propagating layer lower and upper bound activations, it is not possible to scale Cert-R4 to architectures as large as ResNet or BERT, as the results would be trivial. Additionally, this is one of the reasons we introduced Rand-R4, which can scale to arbitrarily large architectures, but is the least tight optimization problem approximation, and Adv-R4, which balances tightness and scalability. We add further clarifications regarding the reason for introducing these techniques at the beginning of Section 4.2 of our revised manuscript.
>
> 3. *“I think the main paper should include some results and discussion regarding the ablations of model architectures[...]”* - The reviewer is right when saying the model size ablations are important to the inference-time performance. However, there is a subtle thing to be noted, namely the fact that the strength of the regularization R4 introduces is kept exactly the same when varying the size of the models, thus the regularization might be too strong or too weak when architecture changes. Notably, since Rand-R4 and Adv-R4 scale up to ResNet and BERT, we believe that the effect of architecture change can be counteracted by the magnitude of $\lambda$ in Eq. 3. The one case in which model size is truly crucial is when making the choice of whether one can employ Cert-R4, which can’t be scaled to large models, as we mentioned in point 2 above. We now clarify the cases when Cert-R4 is applicable and reference the appendix section containing the model size ablation.
>
> 4. We thank the reviewer for their suggestions regarding both the overall polish of the writing and the clarity of the introduction. In response, we have edited the manuscript to improve readability and comprehension. Additionally, we want to clarify to the reviewer that the MLX setting -- the use of masks to annotate shortcut features -- can be approached with any explainability method, i.e. counterfactual or causal explanations.
>
> ## References
> - Gowal, Sven, et al. "On the effectiveness of interval bound propagation for training verifiably robust models." arXiv preprint arXiv:1810.12715 (2018).
> - Heo, Juyeon, et al. "Use perturbations when learning from explanations." Advances in Neural Information Processing Systems 36 (2023): 26872-26897.

---

> > ### Comment · Reviewer_NY1T · 2025-08-13
> >
> > Thank you to the authors for their detailed response to my review.  My major concerns have been addressed, and I support the acceptance of this paper.

---

### Review · Reviewer_Br7p · 2025-08-02

**Summary Of Contributions:**

This paper introduces a simplified, theoretically grounded training objective to mitigate shortcut learning—where models rely on irrelevant features—without depending on unreliable salience-based explanations. The method jointly promotes explanation robustness and discourages shortcut use. Experiments across multiple tasks show a reduction in test-time misclassifications over compared methods. The authors also highlight that annotation quality has a greater impact than quantity.

**Audience:**

Yes

**Broader Impact Concerns:**

This paper aims to improve the robustness of machine learning algorithm, which is beneficial to the community. Thus, no concerns on the ethical implication of this paper.

**Claims And Evidence:**

Yes

**Requested Changes:**

See the weakness section to (1) include proper references (2) define all notations in introduced equations (3) expand the comparison methods in the experiments (4) add reference or experiment for the motivation of proposing the method (5) complete the results in table 1.

**Strengths And Weaknesses:**

***Strengths:***
1. This paper provides theoretical proof on the derivation of the training objectives, which makes the paper more solid.
2. It is a good practice to follow the evaluation protocol in prior study.


***Weakness:***
1. Some claims or arguments in the paper should be supported by appropriate references. For instance, "The best approach to mitigate ... " in Introduction should reference some papers. "Unfortunately, even for ..." also needs reference in Section 3. Also, the notations in Section 4.4.

2. This paper include a number of equations, which also introduce some notations. However, many notations are not defined which makes it hard to follow. For example, $H(f^\theta)$ means Hessian matrix? $sng$ is the signum function? What does $\alpha$ indicate in Section 4.3? If it is a hyperparameter, how is it selected or computed? There are also a few more notations undefined.

3. The proposal of the method is motivated by the observation in Section 4. Given its importance of such a finding, it would be beneficial to introduce some appropriate reference or conduct some experiments to illustrate it.

4. In the experiments, authors only compare the IBP-Ex  $+ R^3$. However, the best performant methods in [a] are IBP-Ex+Grad-Reg and PGD-Ex+Grad-Reg. Please also compare these two methods.

   [a] Use perturbations when learning from explanations.

5. Compared methods are outdated, where the latest method is published in 2023. It may be helpful to include [b].

   [b] Targeted Activation Penalties Help CNNs Ignore Spurious Signals.

6. On Decoy IMDB and Salient Imagenet, why are the performance of some methods missing?

---

> ### Author Response · Authors · 2025-08-13
>
> We thank the reviewer for their thoughtful review of our paper. Below, we address some of the concerns they list:
>
> 1. We thank the reviewer for pointing out notation that is unclear in our paper, in our polishing pass we have gone through and ensured that all notation is properly defined.
>
> 2. *“"Unfortunately, even for ..." also needs reference in Section 3.”* - We point out that this claim is supported by our findings in Table 2. In particular, the reviewer can notice that the $\kappa_m$ (equivalently, $\delta$ or gradient fragility) value outlined in blue is the largest for unregularized models (ERM). In the current revision, we have referenced these results when making the claim.
>
> 3. *“The proposal of the method is motivated by the observation in Section 4. Given its importance of such a finding, it would be beneficial to introduce some appropriate reference or conduct some experiments to illustrate it.”* - We thank the reviewer for pointing out the need for such a crucial observation to be further explained. Indeed, the work of Dombrowski et. al. (2019) shows how the poor robustness or fragility of explanations makes them susceptible to manipulations, hence why previous methods such as R3 are suboptimal in mitigating shortcut learning. We now include references to Dombrowski et. al. (2019) and Dombrowski et. al. (2022) in our revised manuscript.
>
> 4. *“In the experiments, authors only compare the IBP-Ex. However, the best performant methods in [a] are IBP-Ex+Grad-Reg and PGD-Ex+Grad-Reg.”* - In the paper [a], IBP-Ex+Grad-Reg and IBP-Ex+R3 refer to the exact same method, namely IBP-Ex used in conjunction with R3 and weight regularization. As such, the results reported in the paper as IBP-Ex+R3 are obtained using the same method introduced by Heo et al. (2023), which has been shown by the authors to outperform PGD-Ex+R3. We clarify this fact in our revision in the final sentence of Section 5.1.
>
> 5. *“Compared methods are outdated, where the latest method is published in 2023. It may be helpful to include [b].”* - We thank the reviewer for bringing this paper to our attention, we have cited it in our most recent revision. We highlight that the approach by Zhang et al. (2024) is methodologically orthogonal to ours and may represent an interesting future direction of study. Their method uses saliency of intermediate layers of convolutional networks to regularize a model through learned saliency maps. In contrast, our approach regularizes only the input gradient, with given or learned masks, and applies to any architecture. Given these differences in objectives and implementation, we believe a direct comparison between the two approaches would not necessarily be fair. As a minor point, we clarify that two of the methods we compare with (IBP-Ex and IBP-Ex+R3) are those introduced by Heo et al. (2023), a work that is contemporary with the one suggested by the reviewer.
>
> 6. *“On Decoy IMDB and Salient Imagenet, why are the performance of some methods missing?”* - The reason for not reporting the performance of IBP-Ex, IBP-Ex+R3 and Cert-R4 on DecoyIMDB and Salient Imagenet is because all of these techniques employ the Interval Bound Propagation method of Gowal et al. (2018), which, because of its overapproximation errors introduced when propagating worst-case adversarial bounds, cannot scale to large architectures. To further clarify, out of the three methods we introduced for approximating the optimization problem in Eq. 3, Cert-R4 is the tightest, yet least scalable one, Rand-R4 is the loosest, yet most scalable, and Adv-R4 strikes a balance between the previous two. We thank the reviewer for pointing out this concern and we make this point clear in the experimental section of our revision.
>
> ## References
> - Gowal, Sven, et al. "On the effectiveness of interval bound propagation for training verifiably robust models." arXiv preprint arXiv:1810.12715 (2018).
> - Zhang, Dekai, Matt Williams, and Francesca Toni. "Targeted activation penalties help CNNs ignore spurious signals." Proceedings of the AAAI Conference on Artificial Intelligence. Vol. 38. No. 15. 2024.
> - Heo, Juyeon, et al. "Use perturbations when learning from explanations." Advances in Neural Information Processing Systems 36 (2023): 26872-26897.
> - Dombrowski, Ann-Kathrin, et al. "Explanations can be manipulated and geometry is to blame." Advances in neural information processing systems 32 (2019).
> - Dombrowski, Ann-Kathrin, et al. "Towards robust explanations for deep neural networks." Pattern Recognition 121 (2022): 108194.

---

### Review · Reviewer_ZDng · 2025-08-03

**Summary Of Contributions:**

This paper introduces $R^4$, a Machine Learning from Explanations (MLX) framework that promotes robustness in feature attribution by minimizing the model's reliance on irrelevant (non-core) features. Unlike previous methods that primarily focus on robustness to small perturbations in non-core inputs, $R^4$ instead targets robustness of feature importance scores, which helps ensure that explanations remain stable even when the non-core components of an input vary. $R^4$ achieves this by attempting to minimize the impact of the perturbation which would maximally increases attribution to non-core features, which is intractable.  Therefore, the authors propose three approximation strategies that differ in complexity and performance. Rand-$R^4$ approximates the worst case by randomly sampling perturbations and selecting the one with the greatest shift. Adv-$R^4$ uses a first-order gradient-based optimization procedure, similar to adversarial attacks, to find a perturbation that maximizes sensitivity. Cert-$R^4$ instead computes an upper bound on the worst-case attribution using interval bound propagation, allowing for worst-case guarantees. Experimental results across several datasets demonstrate that $R^4$, particularly the Cert-$R^4$ variant, achieves greater robustness in attribution while maintaining strong predictive performance.

**Audience:**

Yes

**Broader Impact Concerns:**

No concerns on the ethical implications of this work.

**Claims And Evidence:**

Yes

**Requested Changes:**

Please refer to the weaknesses part above.

In addition, there are several points where changes are needed:

1. In the left column of Figure 1, it appears that screenshots of plots were used, as indicated by the visible top and left image borders. Cleaning up these borders would improve the visual quality and professionalism of the figure.

2. In the right column of Figure 1, the authors show the performance of their method under corrupted masks. It would strengthen the paper to include how previous methods perform under the same corrupted mask conditions.

3. The limitations section mentions that if multiple shortcuts are present in the data, $R^4$ is believed to handle them effectively and efficiently. However, this claim would be much stronger with supporting empirical evidence. If possible, the authors should include experiments or examples that directly demonstrate this capability.

**Strengths And Weaknesses:**

This paper provides a meaningful contribution to the robustness of MLX approaches, offering a framework that is both mathematically grounded and supported by empirical evidence. The authors conduct extensive experiments demonstrating the robustness of the $R^4$ framework in comparison to previous methods. They also provide thoughtful discussions and explanations that further emphasize the significance of their approach. However, the paper could benefit from a more detailed discussion of the computational requirements of the proposed variants, especially in comparison to prior work. Additionally, it would be interesting to explore how different types of mask corruption affect the performance of earlier methods, which would offer a clearer picture of how Cert-$R^4$ compares to the existing literature under varied conditions.

---

> ### Author Response · Authors · 2025-08-13
>
> We thank the reviewer for the consideration given to our work and address their concerns and comments below.
>
> 1. *“However, the paper could benefit from a more detailed discussion of the computational requirements of the proposed variants, especially in comparison to prior work.”* - The reviewer points out an important point that hasn’t been explained enough, which we added in our revision. To summarize, Cert-R4 leverages the Interval Bound Propagation of Gowal et. al. (2018) and as such, is a strong, but overapproximate regularizer, which can get arbitrarily close to the optimal solutions, but scales poorly on large architectures. In contrast to that, Rand-R4 is very scalable, but is the weakest regularizer among our techniques, managing to get close to optimality only in the limit of samples. Adv-R4 lies between these methods in regards to both scalability and approximation tightness. In terms of computational requirements, Cert-R4 is equivalent to doing two forward and two backward passes, Adv-R4 is equivalent to doing $2 \cdot N$ (where $N$ is user-defined) iterations of the FGSM attack of Goodfellow et. al. (2014), and Rand-R4 is equivalent to doing a forward and backward pass, *per sample*.
>
> 2. *“It would strengthen the paper to include how previous methods perform under the same corrupted mask conditions.”* - We thank the reviewer for suggesting a valuable additional experiment and have added an ablation of R3 and IBP-Ex on DecoyMNIST using the outlined corruption setup, which can be found in Appendix F of our revised manuscript.
>
> 3. *“The limitations section mentions that if multiple shortcuts are present in the data, R4 is believed to handle them effectively and efficiently. However, this claim would be much stronger with supporting empirical evidence.”* - In our current work we clarify our point regarding the existence of multiple shortcuts. Li. et al. (2023) claims that having annotations/masks for only a subset of the shortcut features makes unregularized models rely even more on the shortcut features that do not have corresponding annotations. That is, MLX approaches may perform poorly given errors in the algorithms that perform mask acquisition/segmentation. Our experiments demonstrate that provided the amount of qualitative annotations is large enough, our method is able to successfully mitigate shortcut learning, as can be observed in Table 1. The claim of Li. et al. (2023) can be empirically observed in the SHRINK corruption ablation setup in the right column of Figure 1, because by shrinking the annotations area, we essentially leave several shortcut features unmasked, which leads to a decrease in worst group accuracy, as expected.
> ## References
> - Gowal, Sven, et al. "On the effectiveness of interval bound propagation for training verifiably robust models." arXiv preprint arXiv:1810.12715 (2018).
> - Li, Zhiheng, et al. "A Whac-A-Mole Dilemma: Shortcuts Come in Multiples Where Mitigating One Amplifies Others; 2023." URL https://arxiv.org/abs/2212.04825 (2022): 3.
> - Goodfellow, Ian J., Jonathon Shlens, and Christian Szegedy. "Explaining and harnessing adversarial examples." arXiv preprint arXiv:1412.6572 (2014).

---

> ### Comment · Reviewer_ZDng · 2025-08-13
>
> I appreciate the authors' feedback that addresses my concerns well, and thus I support acceptance of the submission.

---

### Decision · Action_Editor_7He2 · 2025-09-10

**Recommendation:** Accept with minor revision

**Additional Comments:**

The reviewers have discussed several suggestions, mainly regarding the exposition of the paper. I would like to see these incorporated into the text (if they are not already in the new revision). These modifications (in no particular order) include:

(1) Make it clearer early on in the introduction that the paper focuses on gradient-based saliency maps.

(2) Improve the quality of Figure 1.

(3) Discuss the running time of the algorithm and the baselines in a clearer way (this can be helpful, especially when some baselines are missing from some figures due to the prohibitive runtime).

(4) It would be good to provide some insights (even in the limitation section) on when the $R^4$ (or its variant) might not work well.

(5) The reviewers have also suggested some additional baselines/ablations. The authors should include these results in the final version. From the responses, the authors either have already performed these experiments or discuss how the baseline is not relevant. For the former case, please include the results in the paper (in the appendix if necessary,) and in the latter, please discuss in related work why a fair comparison is not possible.

**Audience:**

Yes

**Audience Explanation:**

All reviewers found the topic of machine learning from explanation to be of interest to the TMLR audience. In addition, the specific paper tackles a core shortcoming of a lack of robustness in future attribution, which can be of significant interest to the TMLR audience.

**Claims And Evidence:**

Yes

**Claims Explanation:**

The paper introduces new approaches ($R^4$ and its variants) to encourage robustness in feature attribution. The main claim of the paper is that the approaches reduce test-time misclassifications by 20\% compared to baselines and this claim is clearly supported by the experiments in Section 5.